# A closed translocation channel in the substrate-free AAA+ ClpXP protease diminishes rogue degradation

Alireza Ghanbarpour [1], Steven E. Cohen [1], Xue Fei[1], Laurel F. Kinman [1], Tristan A. Bell[1], Jia Jia Zhang [1], Tania A. Baker [1], Joseph H. Davis [1,2] & Robert T. Sauer [1,2]

AAA+ proteases degrade intracellular proteins in a highly specific manner. *E. coli* ClpXP, for example, relies on a C-terminal ssrA tag or other terminal degron sequences to recognize proteins, which are then unfolded by ClpX and subsequently translocated through its axial channel and into the degradation chamber of ClpP for proteolysis. Prior cryo-EM structures reveal that the ssrA tag initially binds to a ClpX conformation in which the axial channel is closed by a pore-2 loop. Here, we show that substrate-free ClpXP has a nearly identical closed-channel conformation. We destabilize this closed-channel conformation by deleting residues from the ClpX pore-2 loop. Strikingly, open-channel ClpXP variants degrade non-native proteins lacking degrons faster than the parental enzymes in vitro but degraded GFP-ssrA more slowly. When expressed in *E. coli*, these open channel variants behave similarly to the wild-type enzyme in assays of filamentation and phage-Mu plating but resulted in reduced growth phenotypes at elevated temperatures or when cells were exposed to sub-lethal antibiotic concentrations. Thus, channel closure is an important determinant of ClpXP degradation specificity.

AAA+ proteases carry out ATP-dependent protein degradation in all cells[1]. In a series of energy-consuming reactions, a AAA+ ring hexamer binds a target protein, unfolds any native structure present, and then translocates the unfolded polypeptide through an axial channel and into the chamber of an associated protease for degradation. The AAA+ ClpXP protease consists of one or two copies of the ClpX$_6$ unfoldase/translocase and the double-ring ClpP$_{14}$ peptidase[2,3]. *Escherichia coli* ClpXP can degrade hundreds or thousands of distinct cellular substrates tagged with C-terminal ssrA degrons added by tmRNA-mediated ribosome rescue when protein biosynthesis stalls[4,5]. This observation and experiments with unnatural polypeptides[6] suggest that ClpX can translocate almost any amino-acid sequence through its axial channel and into the proteolytic chamber of ClpP. Promiscuous translocation is also consistent with nonspecific contacts in cryo-EM structures between translocating polypeptides and the axial pore

loops of ClpX that spool substrate sequences into ClpP[7,8]. Surprisingly, however, ClpXP does not degrade unfolded proteins lacking degrons at an appreciable rate, even though the unfolded regions of such proteins could potentially bind in its axial channel to begin degradation. Moreover, some AAA+ proteases, such as Lon, do degrade unfolded or partially denatured proteins with little apparent specificity[9,10]. Given the common structural features and mechanisms of AAA+ proteases[11], we were intrigued by this discrepancy.

The *E. coli* ssrA degron ends with an Ala-Ala-$_{COO^-}$ dipeptide[12]. In a cryo-EM structure of a recognition-complex (https://doi.org/10.2210/pdb6WRF/pdb), this Ala-Ala and nearby ssrA-tag residues bind in the upper portion of an axial channel closed by the pore-2 loop of the top subunit in the hexameric spiral of ClpX[13]. In another structure (https://doi.org/10.2210/pdb6WSQ/pdb) from the same cryo-EM data set[13], the ClpX channel opens and six residues of the ssrA tag are translocated to

[1]Department of Biology Massachusetts Institute of Technology Cambridge, Cambridge, MA 02139, USA. [2]These authors contributed equally: Joseph H. Davis, Robert T. Sauer. e-mail: jhdavis@mit.edu; bobsauer@mit.edu

allow subsequent ATP-dependent substrate unfolding, further translocation, and eventual substrate degradation.

These observations led us to ask whether the axial channel of ClpX is closed in the absence of substrate or whether ssrA-tag binding induces or stabilizes channel closure. Here, by determining cryo-EM structures of substrate-free variants of ClpXP, we show that the channel is normally closed. We also demonstrate that structure-guided deletion of residues from the pore-2 loop, which is predicted to destabilize the closed-channel conformation, slows degradation of an ssrA-tagged substrate but increases the rate of degradation of poorly folded substrates lacking degrons. In *E. coli*, the open-channel ClpXP variant results in phenotypes that indicate it can degrade FtsZ and disassemble very stable complexes of the MuA transposase bound to recombined DNA. However, this mutant did not support growth at 42 °C as well as wild-type ClpXP and also resulted in increased sensitivity to kanamycin, an antibiotic that causes amino-acid misincorporation during protein synthesis[5]. In combination, our results show that axial-channel closure affects ClpXP degradation activity and specificity.

## Results and discussion

### Substrate-free ClpXP has a closed translocation channel

We determined three cryo-EM structures of substrate-free variants of *E. coli* ClpXP (Fig. 1; Supplementary Figs. 1–7; and Supplementary Table 1). The common parts of these structures were essentially identical despite differences that included: (i) whether complexes contained full-length ClpX or a variant containing six genetically tethered subunits lacking the N-terminal domain (called single-chain ClpX$^{\Delta N}$);[14] (ii) whether ATP or ATPγS was used to stabilize ClpX binding to ClpP; and (iii) whether the double-ring ClpP tetradecamer was bound by one or two hexamers of ClpX or single-chain ClpX$^{\Delta N}$ (Supplementary Table 1). The substrate-free structure with full-length ClpX represented ~12% of particles in a data set that included a λO-tagged substrate which bound too weakly to saturate all ClpXP. For the doubly capped ClpXP structures, only one of the two ClpX rings was included in the refinement because the asymmetric conformations of the two ClpX hexamers are uncoordinated and thus one has poor resolution as a result of averaging of unaligned subunits within the distal hexamer. The N-terminal domains of ClpX were not visible in the full-length structure (Supplementary Figs. 1, 5, and 6), presumably as a consequence of flexible linkage to the rest of ClpX. Final refined models typically included residues 65-413 of each ClpX subunit, four ATPγS/ATP and two ADP molecules, and residues 2–193 of each ClpP subunit. Residues 192-203 of the pore-2 loop of the lowest ClpP-proximal subunit in the ClpX spiral (chain F) had poor density and were not modeled. The axial portal of the distal ClpP ring in the singly capped structure was closed (Supplementary Fig. 6), whereas the axial portals of ClpP rings contacting ClpX were open.

The substrate-free structures aligned to each other and to the ssrA-degron recognition complex with Cα RMSDs of less than 1 Å. Moreover, the pore-2 loop of ClpX subunit A blocked the translocation channel in the substrate-free structures (Figs. 1 and 2) in the same manner as in the recognition complex[13]. Thus, channel closure preexists in most substrate-free ClpXP complexes and is not induced by binding of the ssrA degron. The closed channel in our substrate-free structures differed from ClpXP structures in which translocating polypeptides were present in an open axial channel and the pore-2 loop of ClpX subunit A was partially disordered[7,8].

### Multistep ssrA-tag binding

The RKH loop of ClpX subunit C in our substrate-free structures assumed a conformation that would clash with a bound ssrA degron (Fig. 3 and Supplementary Fig. 8), suggesting that the tag binds in two steps. Initially, the Ala-Ala-$_{\mathrm{coo}^-}$ of the ssrA tag would bind the pore-1 and/or pore-2 loops of ClpX chains A and B in essentially the same conformation observed in our substrate-free structures, possibly displacing the RKH loop of chain C. In a second step, illustrated in Supplementary Movie 1, downward movement of the RKH loop of subunit C could then stabilize the bound conformation of the ssrA degron seen in the recognition complex.

### Equilibration between closed- and open-channel structures

We hypothesized that open-channel and closed-channel conformations of ClpX could be in equilibrium, with the closed-channel structure predominating in the absence of substrate. If true, only a small fraction of substrate-free particles should adopt an open-channel conformation. To identify such particles and to quantify their relative abundance, we trained a cryoDRGN[15,16] model on our doubly capped single-chain ClpX$^{\Delta N}$/ClpP dataset and used this model to generate density maps corresponding to each particle (see Methods). Manual inspection of the channel in a subset of 250 such maps and automated assessment of the full set of ~100,000 maps provided consistent estimates of the occurrence of open-channel particles, with the automated approach revealing ~4% of particles in an open-channel conformation, ~89% in a closed-channel conformation, and the remainder in a conformation that could not be confidently assigned as open or closed (Supplementary Fig. 9). Additionally, by providing cryoSPARC's heterogeneous refinement tool[17] with exemplar cryoDRGN-generated open and closed maps as initial models, we independently estimated that ~5% of particles adopted an open-channel conformation.

### Destabilizing the closed channel alters ClpXP degradation specificity

To probe the functional role of channel closure, we excised Asn[195]-Pro[196]-Ser[197] from the center of the ClpX pore-2 loop, as modeling

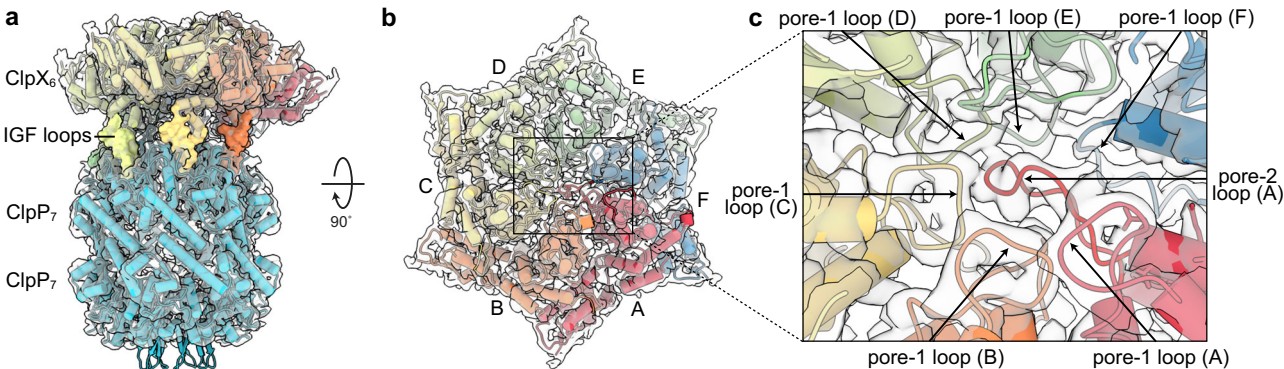

**Fig. 1 | Cryo-EM structure of substrate-free particles containing single-chain ClpX$^{\Delta N}$ and ClpP.** Structural side (**a**) and top (**b**) views. Cryo-EM density is colored light gray and fitted atomic models are displayed in cartoon representation and colored and labeled by ClpX subunit. Key structural features are noted. **c** Closeup of the axial ClpX pore, showing the locations of the six pore-1 loops and the pore-2 loop of subunit A.

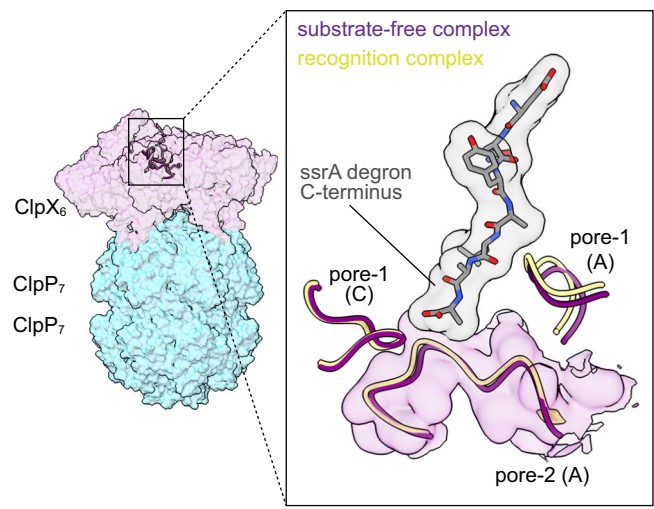

**Fig. 2 | The pore-1 and pore-2 loops of ClpX adopt similar conformations in the substrate-free and recognition complexes.** Side view of the complex of single-chain ClpX$^{\Delta N}$ and ClpP with semi-transparent surface representations of ClpX (purple) and ClpP (cyan). Inset highlights similar conformations of pore-1 (residues 150-156) and pore-2 loops from subunits A and C in the substrate free (purple) and recognition complex (yellow) structures. An atomic model of the ssrA degron from the recognition complex is depicted as a stick model and gray surface representation.

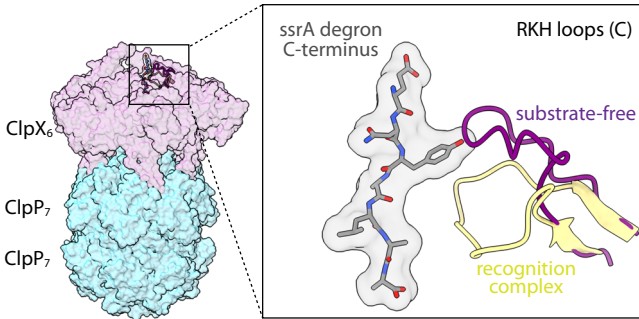

**Fig. 3 | Flexible RKH loops assume altered conformations in the presence and absence of substrate.** The structure on the left shows a semi-transparent surface representation of the substrate-free complex of single-chain ClpX$^{\Delta N}$ and ClpP. The inset highlights the ssrA degron (stick representation with a transparent van der Waal's surface) from the ClpXP recognition complex (https://doi.org/10.2210/pdb6WRF/pdb) and RKH loops of chain C from the substrate-free and recognition structures (residues 220-238; cartoon representation). Note that the aromatic Tyr side chain in the ssrA tag would sterically impinge on the substrate-free conformation (purple) of this RKH loop.

suggested that this deletion (hereafter called ΔNPS) would stabilize the open-channel conformation but maintain wild-type contacts with ClpP. A larger deletion/substitution of the pore-2 loop weakens ClpP binding and prevents degradation of ssrA-tagged substrates[18]. Compared to the parent enzyme, ΔNPS ClpX$^{\Delta N}$/ClpP degraded GFP-G$_3$YG$_9$SENYALAA (called GFP-ssrA*, as the last seven residues are identical to the ssrA tag) substantially more slowly and with an approximate linear rather than hyperbolic dependence on substrate concentration (Fig. 4a). These results indicate that the mutant binds GFP-ssrA* less tightly, thereby increasing $K_M$ for degradation, and support the idea that the ΔNPS mutation destabilizes the closed-channel conformation that normally binds the ssrA tag. Despite this binding defect, the ΔNPS enzyme still unfolded and supported ClpP degradation of the highly stable native domain of GFP. ΔNPS ClpX$^{\Delta N}$/ClpP also degraded GFP-ssrA* less efficiently than did the parental enzyme in *E. coli* (Fig. 4b and Supplementary Fig. 10).

To test degradation of a protein lacking stable tertiary structure, we unfolded untagged or ssrA-tagged variants of the human titin$^{I27}$ domain by modifying cysteines normally buried in the three-dimensional structure with 5-iodoacetomidofluorescene[19], resulting in unfolded substrates called $^{AF}$titin and $^{AF}$titin-ssrA, respectively. As expected, ClpX$^{\Delta N}$ supported faster ClpP degradation of the ssrA-tagged than the untagged unfolded substrate (Fig. 4c). By contrast, ΔNPS ClpX$^{\Delta N}$ supported similar rates of ClpP degradation of the tagged and untagged substrates, which were faster than ClpX$^{\Delta N}$/ClpP degradation of the untagged substrate (Fig. 4c). Hence, shifting the equilibrium to favor the open-channel conformation of ClpX reduces the degron dependence of proteolysis of unfolded $^{AF}$titin.

We also assayed degradation of FITC-casein, a poorly structured molten-globule protein[20]. ΔNPS ClpX$^{\Delta N}$ supported ClpP degradation of this substrate at a rate ~10-fold faster than ClpX$^{\Delta N}$, whereas full-length ΔNPS ClpX supported ClpP degradation at a rate only ~2-fold faster than wild-type ClpX (Fig. 4d). The latter result suggests that FITC-casein binds to the N-terminal domains of full-length ClpX, thereby tethering the substrate above the axial channel and allowing it to engage transiently populated open-channel conformations. Notably, degradation of FITC-casein by ΔNPS ClpX$^{\Delta N}$/ClpP was faster in the presence of ATP than the slowly hydrolyzed ATPγS analog

(Supplementary Fig. 11). Thus, fast ATP hydrolysis is required for rapid degradation of FITC-casein by ΔNPS ClpX$^{\Delta N}$/ClpP, ruling out models in which an unfolded substrate passively diffuses through an open ΔNPS ClpX channel and into ClpP.

## Cellular phenotypes of open-channel ClpX expression

Deletions of *clpX* or *clpP* have little effect on *E. coli* growth or viability except under stress or when other AAA+ proteases are absent[21]. In principle, expression of ΔNPS ClpX in *E. coli* might affect cell physiology by failing to degrade normal ClpXP substrates, such as ssrA-tagged proteins, and/or by degrading new substrates. In two cell-based assays, expression of ΔNPS ClpX had little or no effect compared to wild-type ClpX. For example, plaque formation by bacteriophage Mu depends on ClpX disassembly of a highly stable complex of the MuA transposase with recombined DNA[22]. Mu formed plaques on *E. coli* Δ*clpX* cells expressing ClpX or ΔNPS ClpX but not on cells lacking any ClpX variant (Fig. 5a). Overproduction of wild-type ClpXP results in cell filamentation as a consequence of FtsZ degradation[23]. We found that cells expressing ΔNPS ClpX/ClpP and wild-type ClpX/ClpP formed similar numbers of filaments, whereas filamentation was not observed in cells lacking ClpX (Fig. 5b). Thus, the ΔNPS mutation does not substantially alter the ability of ClpX to mediate disassembly or degradation of some native substrates.

Under some conditions, ΔNPS ClpX did not support cell growth or viability as well as the closed-channel parent. For example, cells expressing ΔNPS ClpX/ClpP grew more slowly than ones expressing wild-type ClpX/ClpP at 42 °C (Fig. 5c), although growth rates at 37 °C were similar (Supplementary Figs. 12 and 13). Moreover, cells expressing ClpX/ClpP showed ~10-fold better recovery than those expressing the ΔNPS variant when plated on sublethal concentrations of the antibiotic kanamycin (Fig. 5d). Because kanamycin causes readthrough of translational stop codons and results in ssrA tagging[24], this phenotype could reflect defective ClpP degradation of ssrA-tagged proteins by the ΔNPS mutant.

## The ClpX channel is not a major route for nonspecific peptide entry into ClpP

To minimize nonspecific degradation, an axial portal in each heptameric ring of ClpP controls access to the degradation chamber, opening when in contact with ClpX but closing in its absence (Supplementary Figs. 4 and 5)[7,8,13,25,26]. Portal opening helps explain why ClpX stimulates ClpP cleavage of degron-free peptides as long as 20 amino acids[27], but our predominantly closed-channel structures

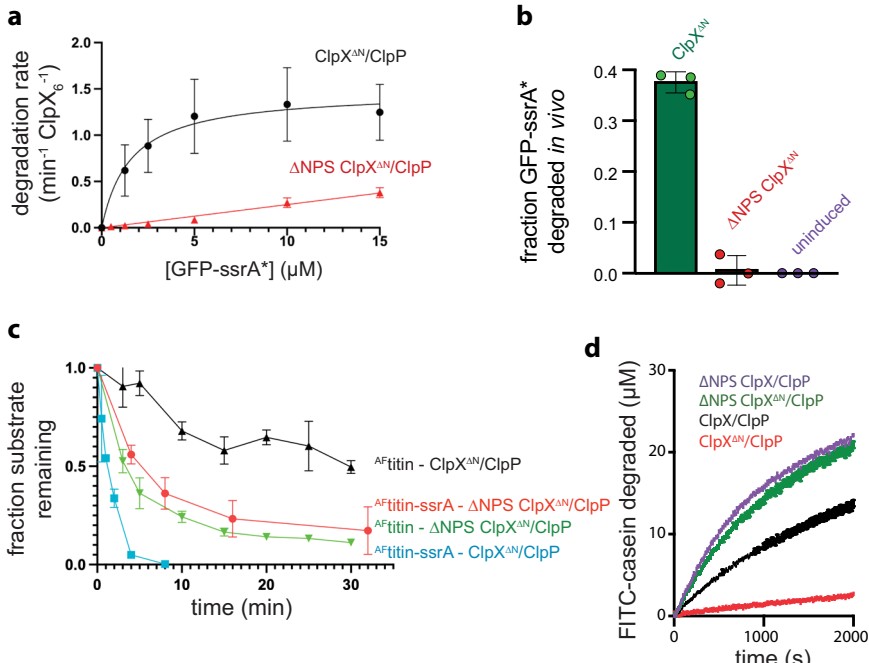

**Fig. 4 | Effects of the ΔNPS ClpX mutation on protein degradation. a** Loss of GFP fluorescence was used to assay the kinetics of ClpP (1.5 µM) degradation of different concentrations of GFP-ssrA* at 30 °C supported by ClpX[ΔN] or ΔNPS ClpX[ΔN] (0.5 µM). Symbols represent means ± 1 SD (*n* = 3 independent experiments). **b** After expression of GFP-ssrA* from a constitutive promoter at 37 °C in *E. coli* T7 Express Δ*clp*AΔ*clp*PΔ*clp*X cells, the bar chart depicts the fraction of GFP-ssrA* degraded 30 min after inducing co-expression of ClpX[ΔN]/ClpP, ΔNPS ClpX[ΔN]/ClpP, or an uninduced control as assessed by fluorescence microscopy. Values are means ±

1 SD (*n* = 3 independent experiments) with individual data points shown as symbols. **c** SDS-PAGE was used to assay degradation of unfolded [AF]titin or [AF]titin-ssrA (10 µM) at 30 °C by ClpX[ΔN] or ΔNPS ClpX[ΔN] (2.5 µM) and ClpP (2.5 µM). Symbols represent the average substrate remaining ± SD (*n* = 3 independent experiments). **d** FITC-casein (50 µM) degradation at 30 °C by purified variants of ClpX or ClpX[ΔN] (0.5 µM) and ClpP (1.5 µM). Data points represent means of three technical replicates. Source data are provided as a Source Data file.

suggest that such peptides enter ClpP by a non-channel route. This model is also supported by the fact that ClpX stimulation of ClpP cleavage of a degron-free decapeptide does not depend on ATP hydrolysis, which is required for peptide translocation through the channel[27].

To test a non-channel entry model, we assayed ClpP cleavage of a nonspecific decapeptide[27] in the presence or absence of a protein substrate (DHFR-GSYLAALAA) that is engaged, unfolded, and translocated through the channel[28]. Notably, the rate of ClpX-stimulated decapeptide cleavage decreased only marginally in the presence of near saturating concentrations of DHFR-GSYLAALAA (Fig. 6a), both when ATP-dependent protein degradation by ClpX[ΔN]/ClpP was ongoing and when degradation was prevented by a Walker-B mutation (E185Q) that impairs ATP hydrolysis but not substrate binding[29]. Moreover, the nonspecific decapeptide was cleaved by open-channel ΔNPS ClpX[ΔN]/ClpP at a rate within error of the ClpX[ΔN]/ClpP rate and much faster than free ClpP (Supplementary Fig. 14). Conversely, the degradation rate of the DHFR substrate was not slowed in the presence of the decapeptide (Fig. 6b).

We conclude that most decapeptides do not pass through the ClpX channel to enter ClpP. Most ClpP was doubly capped in the experiment of Fig. 6, as ClpX[ΔN] was in 20-fold excess of ClpP and above $K_D$ for binding. Moreover, the portal of the distal ClpP ring in any singly capped ClpXP complexes that might have been present would be closed based on our structure (Supplementary Fig. 5c), making entry via this route unlikely.

### Non-channel passage of degron-free peptides into and out of the ClpP chamber
How do degron-free peptides enter ClpP in the ClpX/ClpP complex if they don't traverse the axial channel of ClpX? When ClpX binds ClpP,

IGF loops from six subunits of a ClpX hexamer dock into six of seven hydrophobic clefts on a heptameric ring of ClpP, leaving a gap of ~10 Å between the IGF loops flanking the empty cleft[7,8,13,26] (Fig. 7). Passage of peptides through this IGF-gap and the open portal would allow non-specific peptides to enter the degradation chamber of ClpXP without requiring ATP hydrolysis, as observed[27].

Importantly, microscopic reversibility dictates that if peptides of 10–20 residues can enter the degradation chamber of ClpXP by passing between gapped IGF loops and through the open portal, then peptides of similar size could exit by the same pathway. Indeed, the peptide products of protein degradation are typically 6-28 residues in length[30,31], and the ~30 Å diameter of the open ClpP portal is wide enough to allow peptide-product egress concurrently with translocation of new substrate polypeptide into the chamber for degradation. An alternative model posits that structural fluctuations at the equatorial ring-ring interface of ClpP create transient windows that allow egress of peptide products[32]. However, an IGF-gap is a feature of all structures of ClpX or ClpX[ΔN] bound to ClpP[7,8,13,26], whereas equatorial windows have not been detected structurally.

### Does channel closure regulate specificity in other AAA+ proteases?
If a closed translocation channel of ClpX is biologically important in allowing degradation of some normal substrates and preventing degradation of other proteins, how do other AAA+ proteases deal with the same problem? At low temperature, the axial channel of the *E. coli* HslUV protease is also blocked, albeit by a completely different molecular mechanism. Specifically, regions of a domain unique to the AAA+ HslU hexamer form a trimeric plug that engages and blocks access to the translocation channel[33]. This autoinhibitory plug melts at higher temperatures, activating proteolysis by HslUV. In substrate-free

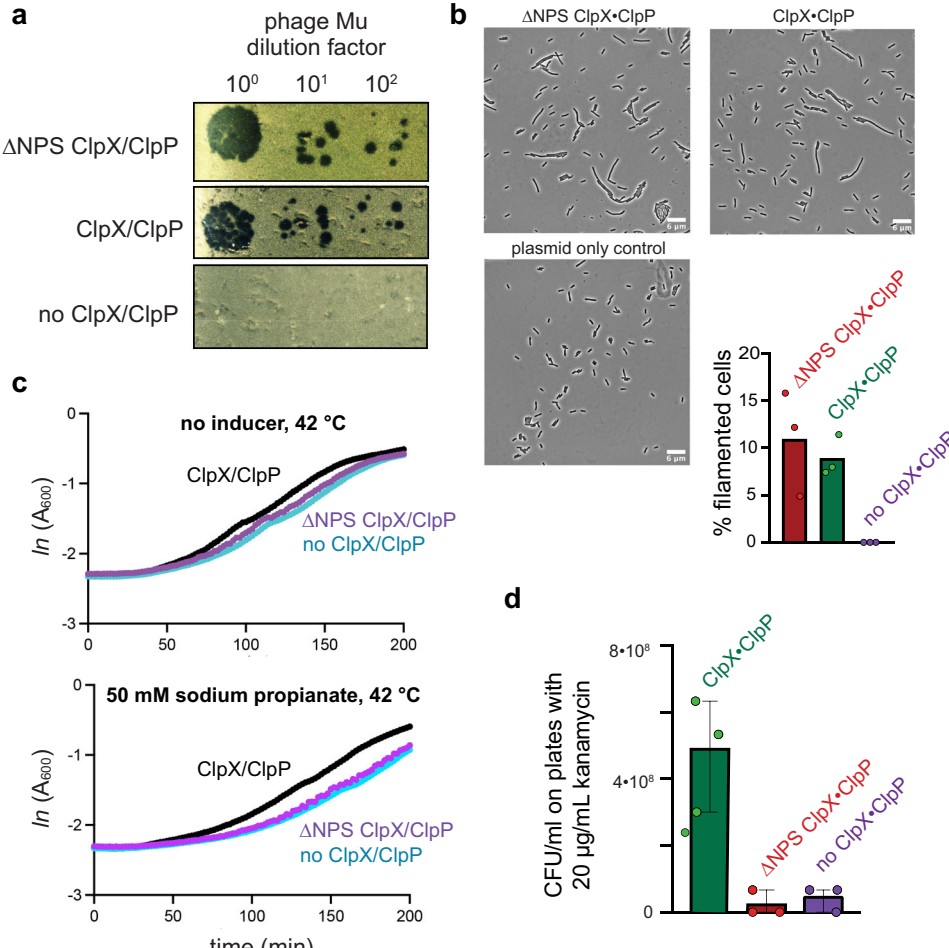

**Fig. 5 | Effects of ΔNPS ClpX on cellular phenotypes.** Assays were performed using *E. coli* W3110 *ΔclpX* cells expressing ClpP and different ClpX variants from a pPro33 plasmid under control of a sodium-propionate inducible promoter[46]. **a** Dilutions of a phage Mu*c^{ts}* stock were spotted onto lawns of cells expressing ΔNPS ClpX/ClpP, ClpX/ClpP, or an empty-vector control (37 °C; 12.5 mM sodium propionate). **b** Micrographs of cells (scale bar 6 μm) expressing ΔNPS ClpX/ClpP, ClpX/ClpP, or an empty-vector control 3 h after induction with 12.5 mM sodium propionate and growth at 37 °C. The graph quantifies the mean number of filamentous cells under these conditions (*n* = 3 technical replicates) with individual data points shown as symbols. **c** Growth curves at 42 °C with and without induction of ClpX variants using 50 mM sodium propionate. Data points are means (*n* = 3 of 3 technical replicates). **d** Equivalent numbers of cells expressing ΔNPS ClpX/ClpP, ClpX/ClpP, or an empty-vector control were applied to individual LB agar plates containing 20 μg/mL kanamycin and 12.5 mM sodium propionate and plates were incubated at 37 °C overnight. The graph shows mean colony forming units (CFU; *n* = 3 independent biological experiments) ± 1 SD with data points shown as symbols, determined by colony counting and correction for the dilution. Source data are provided as a Source Data file.

structures of the human 26S proteasome, the translocation channel of the AAA+ unfolding ring is open but the axial portal into the degradation chamber is typically closed[34], providing another mechanism to limit nonspecific degradation. Substrate-free structures of Lon protease from *E. coli* and *Yersinia pestis* form open lock-washer spirals[35,36], but how these structures bind substrate and transition to the active closed-spiral conformation is unclear.

## Methods

### Protein purification

Variants of *E. coli* ClpX (with a neutral K408E mutation), ClpX^{ΔN} (residues 62-424), single-chain ClpX^{ΔN}, and ClpP contained C-terminal His$_6$ tags and were expressed from plasmids and purified by Ni$^{++}$-NTA, gel-filtration, and ion-exchange chromatography[37,38]. *E. coli* ClpP and full-length ClpX encoded in pT7 plasmids were expressed in *E. coli* strain HMS174(DE3) and purified by Ni$^{++}$-NTA, gel-filtration, and ion-exchange chromatograpy[37]. After protein expression at 30 °C for 3–4 h, cells were harvested by centrifugation at 4500 rpm for 30 min, resuspended in 50 mM Tris HCl (pH 8), 100 mM KCl, 1 mM MgCl$_2$, 5 mM DTT, and 10% glycerol, and lysed using a cell disruptor. Insoluble

material was removed by centrifugation at 25,000 rpm in a Ti45 rotor for 30 min. ClpX in the supernatant fraction was purified by ammonium-sulfate precipitation (35%), phenyl-Sepharose chromatography (Amersham HP 17-1082-01), and chromatography on a self-packed mono-Q ion-exchange column. ClpP was purified using Ni$^{++}$-NTA affinity and size-exclusion chromatography[37]. ΔNPS ClpX^{ΔN} was expressed in T7 Express cells and purified using Ni$^{++}$-NTA and size-exclusion chromatography. N-terminally flag-tagged ΔNPS ClpX was expressed in T7 Express *ΔclpA/ΔclpP/ΔclpX* cells at 18 °C overnight. The protein was bound to M2-FLAG resin (Millipore Sigma Inc., A2220) and eluted with FLAG peptide (APExBIO, Inc.). Purified proteins were concentrated, flash-frozen in liquid nitrogen, and stored at −80 °C.

### Cryo-EM sample preparation

Single-chain ClpX^{ΔN} (3 μM pseudohexamer) and ClpP (1.5 μM tetradecamer) in 20 mM HEPES (pH 7.5), 100 mM KCl, and 25 mM MgCl$_2$ were incubated with ATP (5 mM) for 15 min at room temperature to ensure that any protein substrate present as a consequence of copurification was degraded. Prior to vitrification, samples (2 μL) were placed on 400-mesh Quantifoil 2/1 copper grids, which had been

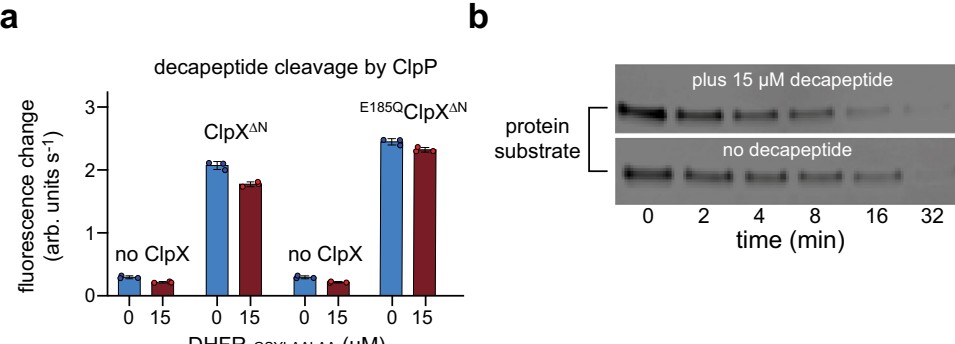

**Fig. 6 | ClpP-mediated decapeptide cleavage. a** Rates of ClpP cleavage of the Abz-KASPVSLGY$^{NO2}$D decapeptide (15 μM) were assayed by increased fluorescence in the presence/absence of ClpX variants (1 μM), ClpP (50 nM), and either 0 or 15 μM DHFR-GSYLAALAA protein substrate. Values are means (*n* = 3 independent experiments) ± 1 SD with data points shown as symbols. **b** Degradation of fluorescent DHFR-GSYLAALAA (15 μM) by ClpX$^{ΔN}$ (0.5 μM) and ClpP (1.5 μM) assayed by SDS-PAGE proceeded at similar rates in the presence and absence of the nonspecific decapeptide (15 μM). Fluorescent DHFR-GSYLAALAA ($M_R$ 18.8 kDa) comigrates with ClpP ($M_R$ 21.5 kDa) on the SDS gel. Source data, including a Coomassie-Blue stained gel for panel **b** and a fluorescent scan of the gel, are provided as a Source Data file.

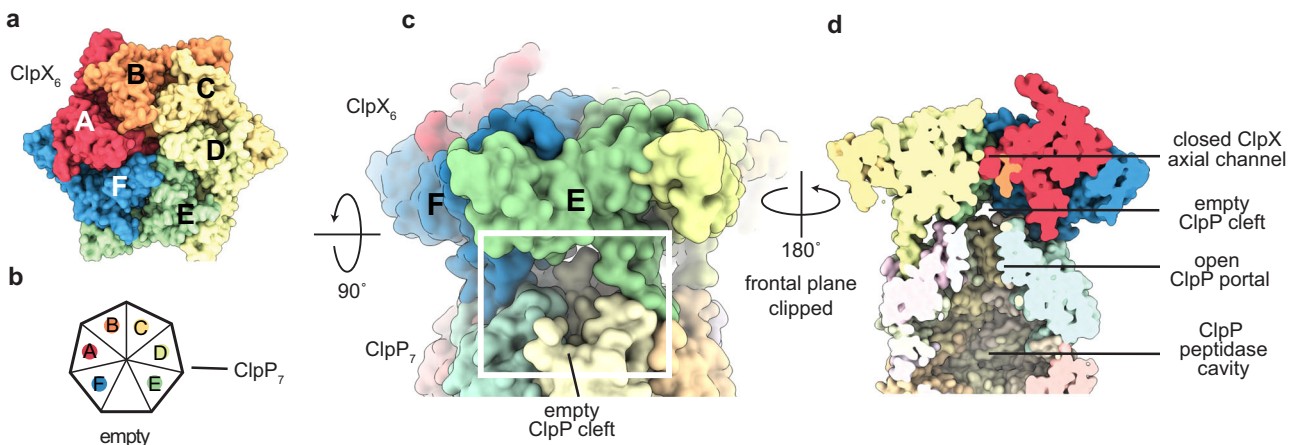

**Fig. 7 | A symmetry mismatch at the ClpX/ClpP interface could allow peptide-product release. a** Top-view surface representation of a ClpX$^{ΔN}$ hexamer bound to ClpP. **b** Cartoon showing how the six IGF loops of ClpX (colored as in panel **a**) dock into six of the seven clefts of a heptameric ClpP ring. **c** Side-view surface representations of the single-chain ClpX$^{ΔN}$/ClpP complex depicting the presence of an empty cleft between the IGF loops of chains E and F. **d** Surface representation clipped in-plane at the ClpXP midline with key structural features noted.

glow-discharged for 60 s in an easiGlow glow discharger (Pelco) at -15 mA, and were blotted using a FEI Vitrobot Mk IV instrument for 4 s with a blot force of +10 (22 °C; 99% relative humidity). The full-length substrate-free structure was obtained from an experiment in which ClpX and ClpP were incubated at room temperature with ATPγS (2.5 mM) for 5 min at room temperature in EM buffer (25 mM HEPES (pH 7.5), 100 mM KCl, and 25 mM MgCl₂), prior to addition of λO-Arc and ATPγS. Final concentrations were ClpX₆ (5.8 μM), ClpP₁₄ (1.9 μM), λO-Arc (20 μM dimer), and ATPγS (2.5 mM). Samples (2.5 μL) were placed on 200-mesh Quantifoil™ R 2/1 copper grids, which had been glow-discharged for 60 s in an easiGlow glow discharger (Pelco) at -15 mA, and were blotted using a FEI Vitrobot Mk IV instrument for 3 s with a blot force of +0 (6 °C; 99% relative humidity) and immediately vitrified in liquid ethane.

## Cryo-EM data collection
For the single-chain ClpX$^{ΔN}$/ClpP structure, 2405 movies were collected with EPU (Thermo Fisher Scientific) on a Titan Krios G3i (Thermo Fisher Scientific), operating at an acceleration voltage of 300 kV and magnification of 105,000 X and detected in super resolution mode on a K3 detector (Gatan) for an effective pixel size of 0.87 Å (0.435 Å super resolution). Movies were collected as 30 frames with a defocus range from −0.75 to −2.5 μm and a total exposure per specimen of 49.39 e⁻/Å². For the ClpX/ClpP structure, 11,719 movies were collected with EPU using aberration-free image shift (AFIS) and hole-clustering method on a Titan Krios G3i with an acceleration voltage of 300 kV and magnification of 130,000X and detected in super-resolution mode on a K3 detector for an effective pixel size of 0.6788 Å (binned by 2). Movies were collected as 40 frames with a defocus range from −0.3 to −1.75 μm and a total exposure per specimen of 52.67 e⁻/Å².

## Cryo-EM pre-processing and particle picking
For the doubly capped single-chain ClpX$^{ΔN}$/ClpP structure, data processing was performed using cryoSPARC (v.3.3.1)[39] and default parameters unless noted. Raw movies (2405) were pre-processed using 'Patch Motion Correction', and 'Patch CTF estimation'. Visual inspection revealed that particles primarily adopted top and side views. Approximately 150 such side views were manually selected with picks centered on the ClpX/ClpP interface and particles were extracted (box size 440 × 440 pixels, down-sampled to 320 × 320 pixels) and averaged into a single class using the 2D classification utility. Using this 2D-class average, we applied the 'Template picker' to 20 micrographs, performed a round of 2D classification on 2254 resulting particles, and

selected two representative side-view classes from the resulting 20 classes. These classes were then used for template picking of the full dataset. 'Exposure curation' was used to eliminate 252 micrographs with poor CTF fits and/or too few particles. Extraction of side views from the remaining 2153 micrographs resulted in 154,554 particles (box size 440 × 440 pixels) that were down sampled to 320 × 320 pixels. A similar workflow was applied to isolate top views, relying on a blob picker (minimum particle diameter 40; max particle diameter 150; use ring blob; and min separation distance 4) instead of the manual picker used in the initial stage, resulting in 237,063 top-view particles. Side-view and top-view particles were subjected to 2D classification and curation, resulting in 140,011 and 227,251 particles, respectively. These were then combined, and the resulting set was subjected to the remove-duplicate-particles utility, resulting in a preliminary stack of 333,774 particles. Preprocessing and particle picking for the singly capped single-chain ClpX$^{\Delta N}$/ClpP structure was performed similarly.

Data processing for the ClpXP structure was performed using cryoSPARC (v.3.3.1)[39] and default parameters unless noted. Raw movies (11,709) were pre-processed using 'Patch Motion Correction' and 'Patch CTF estimation'. 171,780 particles were picked using the blob-picker tool through 1000 random micrographs. Particles were extracted (box size 440 × 440 pixels, Fourier cropped to 256) and averaged using the 2D-classification utility. Using this 2D class average, we then applied the 'Template picker' to the full dataset. Particles were extracted (box size 484 × 484 pixels, Fourier cropped to 320 pixels × 320 pixels). After another round of 2D classification, 1,903,710 particles were selected. The particles and micrographs were subjected to manually curate exposures, extracted (box size 440 × 440 pixels, Fourier cropped to 256 × 256 pixels), and averaged using the 2D-classification utility. 1,801,963 particles were selected as a preliminary stack.

## Ab-initio reconstruction, global refinement, and model building

Ab-initio reconstruction and homogeneous refinement using doubly capped single-chain ClpX$^{\Delta N}$/ClpP particles generated an ~4 Å map of the complex. To better center particles on a single ClpX$^{\Delta N}$/ClpP interface, these particles were re-extracted using translations from this reconstruction and a larger box (initial box 600 pixels; down-sampled to 256 pixels) and subjected to 2-class ab-initio reconstruction, which produced an isolated ClpP volume (146,991 particles) and a clear ClpX$^{\Delta N}$/ClpP volume (175,043 particles). Particles corresponding to the ClpX$^{\Delta N}$/ClpP volume were then refined to a GSFSC resolution of 3.2 Å, and particle alignments were used for a final round of particle re-extraction (initial box 680 pixels; down-sampled to 256 pixels), ab-initio reconstruction, and homogeneous refinement with per-particle defocus optimization, resulting in a GSFSC ~2.9 Å map. Using this map, a mask corresponding to the ClpX and the *cis* ring of ClpP was generated in Chimera X-1.3 and, following a terminal round of duplicate removal, the resulting 171,869 curated, centered particles were subjected to 'Local refinement' in cryoSPARC, resulting in a 3.1 Å GSFCS map focused on ClpX$^{\Delta N}$ and the proximal ClpP ring.

For the singly capped single-chain ClpX$^{\Delta N}$/ClpP structure, after ab-initio reconstruction and homogeneous refinement an ~4 Å map of the complex was obtained. The particles were re-extracted from this reconstruction with a larger box (initial box 600 pixels; down-sampled to 256 pixels) and subjected to 2-class ab-initio reconstruction, which led to an isolated ClpP volume (146,991 particles) and a ClpX$^{\Delta N}$/ClpP volume (175,043 particles). By selecting the latter particles and performing heterogenous refinement against the volumes generated from ab-initio reconstruction, free ClpP particles were removed (2441 particles). The remaining particles that contained ClpX$^{\Delta N}$/ClpP were then subjected to homogenous refinement (2.9 Å GSFSC resolution). Volume and particles from homogenous refinement were centered around ClpP using the 'volume-alignment' tool and extracted with larger box size to allow visualization of full particles (box size

900 × 900 pixels, Fourier cropped to 300). After ab-initio reconstruction (one class) and homogenous refinement a 2.6 Å GSFSC resolution map was obtained. The 'remove-duplicate-particles' utility rejected 23,962 particles. The 'kept' particles were used in a 4-class ab-inito reconstruction. Only one class corresponding to singly capped single-chain ClpX$^{\Delta N}$/ClpP was selected. Further homogenous refinement of these 49,824 particles yielded a map at 3.1 Å GSFSC resolution (after FSC-mask auto tightening).

For the full-length ClpX/ClpP structure, ab-initio reconstruction was performed using four classes. One class containing the majority of particles (952,541) was selected for homogenous refinement; the remaining classes represented junk particles, free ClpP, or a low-resolution complex. After homogenous refinement, particles and the map were recentered around ClpX. After another round of homogenous refinement, heterogenous refinement was performed using four classes. Two classes that contained the majority of particles (325,578 and 527,257) were selected for further homogenous refinement. The other classes yielded low-resolution structures. The substrate-free map was obtained by analyzing 527,257 particle stacks using heterogenous refinement with three classes, and further homogenous and local refinement (by employing a mask focused on ClpX and the *cis* ClpP ring; 214,397 particles). The final substrate-free ClpXP map had a GSFSC of ~2.6 Å after FSC-mask auto tightening. Each of the final maps were rescaled using calibrated pixel sizes (0.654 Å and 0.416 Å for × 130,000 and × 105,000 magnifications) based on apo-ferritin data using ChimeraX-1.3[40].

Model building was performed using a combination of ChimeraX-1.3[40], Coot (0.9.4)[41], and Phenix (1.14)[42]. Local resolution was estimated by cryoSPARC implementation of monoRes[43].

## CryoDRGN analysis

The full set of 171,869 particles from the single-chain ClpX$^{\Delta N}$/ClpP dataset was used to train an eight-dimensional latent-variable model in cryoDRGN v0.3.2b[15,16], using a down-sampled box size of 128 (2.21 Å/pixel) and 256 × 3 encoder and decoder architectures. The poses for cryoDRGN training were supplied from the single-chain ClpX$^{\Delta N}$/ClpP consensus reconstruction from cryoSPARC. Prior to cryoDRGN training, a mask corresponding to ClpX was applied to each of the input particle images by integrating the cryoSPARC refinement mask along the relevant projection axis, dilating the particle edge, applying a falling cosine edge to the resulting image mask, and then multiplying the input image against this mask. Using this trained model, residual junk particles were removed by manually inspecting volumes sampled via *k*-means clustering of the latent space and removing particle corresponding to poorly resolved reconstructions, resulting in a stack of 96,539 particles. A new cryoDRGN model was trained on the filtered dataset, using the full box size of 256 (1.11 Å/pixel), an eight-dimensional latent variable, and 1024 × 3 encoder and decoder architectures. A final filtering was performed on the resulting set of particles by eliminating all particles from consideration with latent encodings of UMAP1 < 1.8 or UMAP2 > 7.6, producing a final set of 93,663 particles.

Each particle in the final stack was queried for ClpX pore-2 occupancy at a down-sampled box size of 64 (4.42 Å/pixel) using the on-the-fly reconstruction and analysis approach implemented in MAVEn[16,44]. A mask for pore-2 residues 192-199 of ClpX chain A was created using alpha carbons from the single-chain ClpX$^{\Delta N}$/ClpP atomic model (https://doi.org/10.2210/pdb8E8Q/pdb) and this mask was used to calculate the pore-2 density occupancy for each reconstructed map. Each pore-2 occupancy was normalized by the occupancy of a mask encompassing all chains of hexameric ClpX, with the exception of pore-2 residues from chain A. The resulting normalized pore-2 occupancies were used for sampling 27 subsets of particles for homogeneous refinement in cryoSPARC as follows: for a range of values of *N*, starting at 5000 and incrementally increasing until *N* was equal to the size of the full final particle stack, 5000 particles were randomly

sampled without replacement from the $N$ particles with the lowest normalized pore-2 occupancies. Each of these homogeneous refinements was provided the consensus reconstruction from the full dataset (low-pass filtered to 30 Å) as an initial model. The resulting 27 maps were queried for normalized pore-2 occupancy at full resolution as described above (Supplementary Fig. 9a). The observed positive correlation between $N$ and measured pore-2 occupancy in the high-resolution maps indicated that the low-resolution on-the-fly MAVEn analysis effectively sorted particles based on pore-2 occupancy.

To calculate the number of particles likely bearing an open channel, $k$-means clustering was performed on the 12,500 particles with the lowest normalized pore-2 occupancies as assessed by on-the-fly MAVEn analysis, and the cluster-center volumes were reconstructed at box size 256 using the cryoDRGN decoder. The resulting 250 density maps were labeled open channel or closed channel by expert-guided manual inspection/assignment or by MAVEn calculation of normalized pore-2 occupancy (Supplementary Fig. 9b). In the latter approach, all volumes with a normalized pore-2 occupancy <0.0015 were labeled open channel, all volumes with a normalized pore 2 occupancy greater than 0.002 were labeled closed channel, and all volumes in between were labeled ambiguous. A high correlation between open and closed annotations from manual inspection or MAVEn analysis provided confidence in the automated MAVEn approach. Final estimates for the frequency of open-channel and closed-channel particles in the dataset used MAVEn assignments, assigning all particles within a cluster the same label as the cluster-center volume, resulting in 4139 open-channel particles, 83,275 closed-channel particles, and 6249 ambiguous particles.

## Degradation assays

Casein degradation was performed at 30 °C using 50 µM FITC-casein (C0528; Sigma Aldrich); 1.5 µM of ClpP, and 0.5 µM of ClpX or variants in 25 mM HEPES (pH 7.5), 100 mM KCl, 10 mM MgCl₂, 1 mM DTT, 4 mM ATP, 5 mM phosphocreatine, and 0.05 mg/ml creatine kinase. For studies of FITC-casein degradation using ATPγS, phosphocreatine and creatine kinase were omitted from the reaction. Concentrations of FITC-casein were measured using a NanoDrop spectrometer (ThermoFischer Scientific) at an absorbance of 280 nm ($\varepsilon = 11,460\ M^{-1}\ cm^{-1}$), and the concentrations of ClpX variants and ClpP were calculated using extinction coefficients obtained from https://web.expasy.org/protparam. All components except casein were initially mixed at 30 °C in a 384-well assay plate (Corning, 3575) for 15 min. FITC-casein, which had also been preincubated at 30 °C, was then added to initiate degradation, which was monitored by increases in fluorescence (excitation 360 nm; emission 525 nm with a cutoff of 515 nm) using a SpectraMax M5 plate reader. After degradation ceased, ~20 µg trypsin was added to each well and the reaction was monitored to determine the maximum fluorescence signal. After background subtraction, the RFU was converted to µM FITC-casein at each time point, using the maximum fluorescence signal and calculated casein concentration.

GFP-ssrA* degradation was assayed by decreases in fluorescence (excitation 467 nm; emission 511 nm) using a SpectraMax M5 plate reader. Assays were conducted at 30 °C in buffer containing 25 mM HEPES (pH 7.5), 5 mM MgCl₂, 200 mM KCl, and 10% glycerol, with 0.5 µM ClpX₆ or variants, 1.5 µM ClpP₁₄, 5 mM ATP, 32 mM creatine phosphate, and 0.08 mg/mL creatine kinase.

The human titin$^{I27}$ domain was expressed and purified by Ni$^{++}$-NTA chromatography[19], denatured by adding 10 volumes of 100 mM sodium phosphate (pH 7.2), 2.5 mM EDTA, and 6 M GdnHCl, followed by addition of 15 molar equivalents of DMF-solubilized 5-iodoacetamidofluorescein (5-IAF; Invitrogen, lot 1792368). After incubation for ~2 h at room temperature, excess 5-IAF was reacted with 20 mM DTT. The modified protein was buffer-exchanged into 25 mM HEPES (pH 7.5), 300 mM KCl, 10 mM MgCl₂, 10% glycerol, and 1 mM DTT using a PD-10 desalting column. The final $^{AF}$titin$^{I27}$ concentration was determined by SDS-PAGE analysis using Coomassie staining with BSA

standards. Degradation of $^{AF}$titin$^{I27}$ (10 µM) by 2 µM of ClpX₆ or variants and 2 µM of ClpP₁₄ was assayed at 30 °C in 25 mM HEPES (pH 7.5), 200 mM KCl, 5 mM MgCl₂, and 1 mM DTT, supplemented with 4 mM ATP, 5 mM phosphocreatine, and 0.05 mg/mL creatine kinase. Degradation was monitored by SDS-PAGE on 4-15% Mini-PROTEAN® TGX™ precast protein gels (BIO-RAD) followed by Coomassie staining and imaging with a Bio-RAD instrument.

Cleavage of a synthetic Abz-KASPVSLGY$^{NO2}$D decapeptide (where Abz is a 2-aminobenzoic acid fluorophore and Y$^{NO2}$ is a 3-nitrotyrosine quencher)[27] by ClpP₁₄ (50 nM) was assayed in the presence or absence of ClpX$^{ΔN}$, $^{E185Q}$ClpX$^{ΔN}$, or ΔNPS ClpX$^{ΔN}$ (1 µM hexamer) with or without 15 µM E. coli $^{D132C/C152S}$DHFR-GSYLAALAA with an N-terminal H₆ and AviTag sequence. The fluorescence of this decapeptide (excitation 320 nm; emission 420 nm) increases following ClpP cleavage. Assays were performed at 37 °C in buffer containing 25 mM HEPES (pH 7.6), 100 mM KCl, 20 mM MgCl₂, 1 mM EDTA, 10% glycerol, 5 mM ATP, 32 mM creatine phosphate (Roche), and 0.08 mg/mL creatine kinase (Millipore-Sigma).

To observe the effect of Abz-KASPVSLGY$^{NO2}$D on ClpX/ClpP degradation of DHFR-GSYLAALAA with D132C and C152S mutations, which comigrates with ClpP on SDS-PAGE, the protein was labeled with 15 molar equivalents of DMF-solubilized 5-IAF for 1 h at room temperature. Excess 5-IAF was quenched by incubation with 20 mM DTT for 10 min, followed by buffer exchange into 25 mM HEPES (pH 7.5), 300 mM KCl, 10% glycerol, and 1 mM DTT using a PD-10 column. The final protein concentration was determined by Coomassie staining of SDS-PAGE gels using a BSA standard. Degradation of 15 µM DHFR-GSYLAALAA by ClpX$^{ΔN}$ (0.5 µM) and ClpP (1.5 µM) plus or minus 15 mM Abz-KASPVSLGY$^{NO2}$D was assayed in 25 mM HEPES (pH 7.5), 200 mM KCl, 5 mM MgCl₂, and 1 mM DTT, supplemented with 4 mM ATP, 5 mM phosphocreatine, and 0.05 mg/mL creatine kinase, and monitored by SDS-PAGE on 4-15% Mini-PROTEAN® TGX™ precast protein gels (BIO-RAD) and imaged using a Typhoon FLA7000 (GE Healthcare).

## GFP-ssrA* degradation in vivo

GFP degradation in vivo was assayed by loss of flourescence[38]. Initially, a plasmid expressing GFP-G₃YG₉SENYALAA (GFP-ssrA*) from a constitutive ProD promoter[45] was transformed into T7 Express ΔclpA/ΔclpP/ΔclpX cells carrying empty pBAD18, pBAD18 (ClpX$^{ΔN}$/ClpP), or pBAD18 (ΔNPS ClpX$^{ΔN}$/ClpP). Following overnight growth, colonies were picked and grown in fresh media supplemented with antibiotics and 2% glucose at 37 °C to an A₆₀₀ of -1.2. Samples were centrifuged, and resuspended to an A₆₀₀ of ~0.4 into fresh media containing antibiotic and 80 mM L-arabinose inducer. After 30 min of induction, cells were transferred to a clear-bottom black 96-well plate, and both GFP fluorescence and optical density were measured using a SpectraMax M5 plate reader. To assess the fraction of degraded GFP, the GFP fluorescence was normalized to the cell optical density and compared to the fluorescence of pBAD18 (ClpX$^{ΔN}$/ClpP) or pBAD18 (ΔNPS ClpX$^{ΔN}$/ClpP) and ProD-GFP-G₃YG₉SENYALAA.

## Cellular assays

Cellular assays were conducted in an E. coli W3110 ΔclpX strain containing plasmids in which genes for clpP and clpX or ΔNPS clpX were cloned into a multiple cloning site under control of a propionate-inducible promoter[46]. To generate a polycistronic expression construct, a ribosome binding site (5′-CAAGGAGAATAACG-3′) was added downstream of the clpP stop codon, allowing coordinated expression of ClpP and wild-type or ΔNPS ClpX. Cloned plasmids or an empty vector control were transformed into E. coli W3110 ΔclpX and assayed as described below.

For microscopy, cells were spotted onto a 1.5% agarose pad, allowed to dry, and a 22 × 60 mm coverslip was placed on top of the pad. Phase-contrast and fluorescence images were taken using a Hamamatsu Orca Flash 4.0 camera on a Zeiss Observer Z1 microscope

with a x100/1.4 oil immersion objective. Fluorescence was generated with an LED-based Colibri illumination system. MetaMorph software (Molecular Devices) was used for image acquisition. Fiji (https://www.imperial.ac.uk/medicine/facility-for-imaging-by-light-microscopy/software/fiji/) was used for image analysis. Illumination and brightness settings were identical for all samples within a given experiment.

For phage-Mu experiments, a Mu$^{cts}$ lysogen was grown overnight at 30 °C, and 1 mL of the overnight culture was transferred to 25 mL of fresh LB media and grown to $A_{600}$ of ~1 at 30 °C. Cells were then heat shocked at 45 °C for 15 min and incubated at 37 °C until lysis after ~45 min. After centrifugation at 4 °C for 30 min at 5000 rpm, the supernatant was collected, a few drops of chloroform were added, and the mixture was stored at 4 °C. To create a bacterial lawn, one volume of cell culture ($A_{600}$ ~ 1.7) was mixed with 200 volumes of melted 0.5% LB agar with or without sodium propionate (50 mM) and chloramphenicol (35 µg/mL). Lawns were overlaid on 1.2% LB agar plates with chloramphenicol and/or sodium propionate. Ten-fold serial dilutions of bacteriophage Mu were then spotted on lawns, and plates were incubated overnight at 37 °C to allow plaque formation.

For growth assays, overnight cultures were grown at 37 °C and then diluted to an $A_{600}$ of ~0.05-1 in LB medium containing chloramphenicol (35 µg/mL). 100 µL of bacterial suspensions containing different ClpX variants and ClpP or an empty vector were then transferred to a 96-well clear flat bottom microplate (Corning Inc.). On a separate microplate, different concentrations of sodium propionate (0-50 mM) were prepared, and 100 µL of each sample was transferred to the bacterial plate using a multichannel pipette. The $A_{600}$ was measured every 2 min using a SpectraMax M2 plate reader.

For kanamycin-sensitivity experiments, overnight cultures expressing wild-type ClpXP, the ΔNPS mutant, or an empty plasmid were grown at 37 °C in LB broth with chloramphenicol (35 µg/ml) and sodium propionate (50 mM). Cultures were then diluted in the same broth, grown to an $A_{600}$ of 0.7, six serial 10-fold dilutions were prepared, and 30 µl of the final dilution was spread onto LB agar plates supplemented with sodium propionate (25 mM), chloramphenicol (35 µg/ml), and kanamycin (20 µg/ml). After overnight incubation at 37 °C, plates were imaged to count the number of bacterial colonies.

For filamentation assays, cells were prepared as described for the kanamycin experiment, grown to an $A_{600}$ of ~0.7 at 37 °C and spotted onto a 1.5% agarose pad. Once the spot was dry, a 22 mm × 60 mm coverslip was placed on top of the agarose pad and phase-contrast images were taken using a Hamamatsu Orca Flash 4.0 camera on a Zeiss Observer Z1 microscope using a ×100/1.4 oil immersion objective.

### Reporting summary

Further information on research design is available in the Nature Portfolio Reporting Summary linked to this article.

## Data availability

Structures have been deposited in the Protein Data Bank with identification codes 8E91, 8E8Q (https://doi.org/10.2210/pdb68E9Q/pdb), and 8E7V (https://doi.org/10.2210/pdb6YNV/pdb), and the corresponding maps have been deposited in the Electron Microscopy Data Bank with identification codes 27952, 27946, and 27941. Source data for biochemical and cellular experiments are provided with this paper. Strains and constructs will be provided upon request. Source data are provided with this paper.

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

## Acknowledgements

Supported by NIH grants R01-GM144542 (JHD), R35-GM141517 (RTS), 5T32-GM007287, and NSF-CAREER grant 2046778 (JHD). We thank Daniel Saxton, Mike Laub, and Barrett Powell for assistance and advice. EM samples were prepared at the Automated Cryogenic Electron Microscopy Facility in MIT.nano and screened on a Talos Arctica microscope, which was a gift from the Arnold and Mabel Beckman Foundation.

## Author contributions

A.G. and T.A.Bell purified proteins and performed biochemical assays. A.G., S.E.C. and X.F. prepared samples for EM imaging and collected data. A.G. and J.H.D. processed EM data and performed reconstruction and refinement. A.G. and R.T.S. built and refined models. A.G. performed cellular assays with help from J.J.Z.; A.G. and L.F.K. performed cryoDRGN analysis. A.G., J.H.D. and R.T.S. wrote the manuscript and made figures. T.A.Baker, J.H.D. and R.T.S. supervised the research.

## Competing interests

The authors declare no competing interests.
