## [Peer Review File · Nature Communications]

REVIEWER COMMENTS

Reviewer #1 (Remarks to the Author):

The authors describe cryo-EM structures of the bacterial ClpXP ATP-dependent protease in the substrate-free state. These structures show that the axial pore of the ClpX ATPase is closed in the absence of substrate, despite the entry pore of ClpP compartmental protease being open and ready to accommodate substrate. Specifically, elements in the pore-2 loop of a ClpX subunit positioned low in the ATPase spiral blocks passage of substrates that are presumably displaced to initiate translocation. The authors note that the closed ClpX pore is inconsistent with previous experimental evidence showing nonspecific entry of a short reporter peptide into ClpP upon ClpX binding. Competition assays show that saturation of the specifically degraded *ssrA*-tagged substrates does not significantly impact peptide entry, suggesting alternative entry paths for the two types of substrates. Instead, the authors suggest that these peptides, and short peptide products produced during proteolysis, move via spaces created between the ClpX and ClpP domains.

The substrate-free ClpXP cryo-EM structures are a useful addition to the collection of physiologically relevant states of this protease and the observation that pore-loop 2 of ClpX blocks entry to ClpP in the absence of substrate provides a nice structural explanation for the tight regulation observed for this protease. The microscopy is well-performed and the structures will be of interest to the broader field of regulated proteolysis.

The question of how product peptides are released from compartmental proteases is important and understudied. However, the authors provide only a small amount of experimental evidence to support their model that peptides can be released between the ClpX and ClpP interface. A more detailed investigation is warranted to address the details of this model, such as the size of product that can be efficiently released and the coordination of the ATPase cycle with the appearance of the channels.

Reviewer #2 (Remarks to the Author):

The bacterial AAA+ protease ClpX/ClpP degrades proteins harboring specific degrons like the *SsrA*-tag. Unspecific degradation of other proteins has to be prevented to avoid cellular toxicity. Here the authors present the cryo-EM structure of a substrate-free ClpX/ClpP complex. They show that the pore-2 loop of one subunit blocks the central translocation channel. The new structure represents an extension of former cryo-EM analysis by the authors. A similar structure was reported before in presence of substrate and was termed recognition complex, representing an initial step of substrate engagement. Deleting the pore-2 loop strongly enhanced degradation of a non-specific substrate (FITC-casein) by ClpX/ClpP, suggesting that channel closure contributes to substrate selectivity. This is an interesting finding that requires further support by controls as detailed below. Furthermore, it would be interesting to analyze the physiological consequences of such mutant on e.g. the ClpX/ClpP substrate spectrum. The authors also suggest a new path for the release of peptides generated in the chamber of the associated ClpP peptidase barrel. Former studies (Sprangers et al., 2005) implied that peptide product release happens through the interface between the two heptameric ClpP rings. Accordingly, restricting ClpP mobility at this site was shown before to slow down product release. Here, the authors propose an alternative pathway, namely peptide entry and exit via the asymmetric interface between ClpX and

ClpP. This model is indirectly supported by the finding that ClpX stimulates the degradation of a nonspecific decapeptide compared to peptide degradation by ClpP only even in the presence of the competing specific ClpX-substrate DHFR-SsrA.

The study comes up with two interesting mechanistic aspects, which are important for the understanding of AAA+ proteases. The biochemical support is, however, so far largely indirect and additional controls as detailed below are required to bolster the models. Furthermore, the physiological consequences of a ClpX mutant with reduced substrate selectivity should be explored. Such data could would underline the relevance of channel closure for bacterial homeostasis. The study is therefore send back for major revision.

Major points:

Channel closure is also seen for the initial step of engagement of a specific SsrA-tagged substrate. This indicates that the binding of a specific substrate induces channel opening in a second step. Is it clear why only specific substrate but not unspecific ones trigger this step? The observation that a pore2-loop mutant degrades FITC-casein indicates that unspecific substrates can engage the upper part of the ClpX channel, yet without triggering channel opening.

Figure 4: The pore2-loop mutant was only analyzed in a ClpX deletion construct lacking the N-terminal domain. The same mutant needs to be analyzed in ClpX WT, as NTDs can function as selectivity barriers. Furthermore, the authors need to determine the proteolytic activity of the mutant towards a specific substrate, e.g. harboring an SsrA-tag. Unaltered degradation of such substrate would document that the pore2-loop deletion only affects degradation of unspecific but not specific substrates, as predicted by the model.

The observation that a pore2-loop mutant exhibits reduced substrate selectivity raises the question whether its production causes toxicity in E. coli cells. A reduced cell growth or viability upon mutant expression would indicate a physiological need for the proposed mechanism.

Figure 5: The authors need to document that DHFR-SsrA is degraded at normal rates in presence of unspecific peptide, supporting the idea that the two substrates use different ways to enter the ClpP degradation chamber. So far the authors only assume unaltered DHFR-SsrA degradation but that needs to be documented. The conclusion that the decapeptide does not travel through the ClpX translocation channel is therefore not demonstrated. It is also strongly recommended to use the generated pore2-loop mutant, which leads to enhanced FITC-casein degradation, in the peptide degradation assay. If the model of the authors is correct this ClpX mutant should not affect the rate of peptide degradation, as the peptides are predicted to not travel through the ClpX channel.

Page 5, left column: ClpA/ClpP efficiently degrades casein, leading to the suggestion that the axial channel of ClpA is open in absence of substrates. This conclusion ignores the possibility that ClpA has a higher affinity to casein as compared to ClpX. A binding site for disordered proteins like casein has been described in the ClpB/Hsp104 N-terminal domain, which is homologous to ClpA. Thus differences in substrate binding but not channel opening/closure might underlie the diverse proteolytic activities.

Reviewer #3 (Remarks to the Author):

This manuscript describes the substrate-free structure of ClpXP and reveals a closed conformation of the ClpX ring that was seen previously in the presence of substrate perched on the edge of the axial pore. This new structure verifies that the aforementioned substrate was indeed incumbent to an open pore and not involved in the pore restriction per se. It also confirms the mechanism of pore-loop2-mediated autoinhibition of ClpX-dependent access to ClpP. The authors go on to hypothesize that the non-ClpX-dependent substrates (such as decapeptides) may not travel through the ClpX pore, but instead travel through the gap left by the asymmetric pairing between hexameric ClpX and Heptameric ClpP at the ClpXP interface. While this hypothesis has merit, we could use a few details.

1. How big is the IGF opening within the static EM structure?

2. How do the interactions of the ClpX-pore2 and ClpP-N-terminal loops play into this trans axial access? Figure 1 of reference #24 suggests these loop interactions if not block exterior access, certainly funnel, peptides from ClpX to ClpP. How is trans-axial access impeded or facilitated by these loop interactions?

3. If the pore2 loops make protein:protein contacts, then the increase in activity in figure 4 would be from both release of axial and trans-axial blockage?

4. How is this hypothesis effected by the knowledge that related AAA+ ATPase unfoldases can harbor more than one peptide through the pore? (Lee et al. JBC (2002) and Han eLife (2019))

It is interesting to note that the methods say the "substrate-free" structure was obtained in the presence of substrate λ O-Arc – not observed. While unfortunate luck and heterogeneous analysis of CryoEM particles can easily provide rationale for the substrate-free structure, it seems this tidbit of information is somewhat hidden in the methods details.

1. Should details be provided as to the peptide sequence or reference to its use as a substrate?

The protein preparation, CryoEM SP data collection, processing and model refinement all look appropriate for this work.

Reviewer
Heidi L Schubert

Reviewer #4 (Remarks to the Author):

Ghanbarpour et al. addresses an important question in the regulation of the ClpXP protease, which confers specificity for substrate degradation: Is the ClpX channel opened or closed prior to the binding and translocation of a substrate? In this work, the authors addressed this elegantly with a series of cryo-EM and biochemical experiments. First, they determined cryo-EM structures of a substrate-free full-length and an N-terminally truncated ClpX complexed with ClpP. The cryo-EM structures revealed an open axial ClpP channel and a closed ClpX channel. Second, the authors showed that mutations in the pore loop of ClpX de-regulate ClpP's substrate specificity and degradation. Finally, through biochemical experiments, the authors propose a model for short peptides' entry into the degradation chamber of ClpP through the gap generated by the symmetry mismatch at the interface between ClpX and ClpP.

The manuscript is well-written and addresses a fundamental pathway on how AAA+ proteases carry out degradation of substrates; in particular how ClpXP regulates substrate entry into the degradation chamber. The model is well-supported by the set of experiments performed here. I only have minor suggestions/questions before the publication of this manuscript.

Comments:

- On page 2, the flexible ~60 residues in the N-terminal region of ClpX is mentioned in the text with reference to supplementary figures (Fig S4 and S5). It is not clear where this region would be in the structure. A dashed line or perhaps a schematic would help.
- In the EM images of the full length ClpXP complex, there seem to be 'some' stacking of ClpXP particles (Fig S2). This is not the case in the images with the N-terminally truncated ClpX mutant (Fig S1). Perhaps it is not specific, but did the authors perform focused 3D classification/variability analysis to rule out the possibility that this mediated via N-terminal interactions as previously reported? The box size used here is small, maybe refining/classifying with a larger box size?
- CryDrng2 was used to investigate whether the ClpX channel could be opened by stochastic ATP hydrolysis in the substrate free form. This software is a powerful tool and the results for this particular sample could be shown in the supplementary. Please include some examples of the obtained 3D volumes, in particular the opened channel state.
- The authors propose a model for small peptides' entry through gaps generated by unoccupied pockets due to symmetry mismatch. This model is plausible and would indicate that the power stroke generated by ATP hydrolysis in ClpX is not required for translocation of small peptides. To validate this, did the authors test peptide degradation in the presence of ATPγS or ADP, where power strokes are inhibited?
- In the text, it is indicated that two of the ClpX protomers were in the ADP state in the cryo-EM maps. Are these the two subunits that break the 6-fold symmetry of the hexamer i.e., as observed in the recognition complex? It would be easier to assess this better with corresponding EM densities/coordinates for the bound ATP/ADP molecules in the supplementary materials, showing closeups of the conformational changes in the interface between ClpX protomers.
- A schematic for the proposed model addressing the two-step substrate recognition/engagement would be helpful for the readership.
- Could the authors comment on the ClpX concentration in the cell? Is it higher than ClpP? One would assume that ClpP is more abundant and thus ClpXP is likely to be a single-capped complex in the cell.
- It is difficult to envision how peptide products, to access the equatorial ring-ring gaps, will have to first pass through the same axial pore of ClpP as substrate is being translocated. This is a tight space and would likely affect the rate of degradation. Is it plausible that these peptides exit ClpP through the uncapped end instead? i.e., the axial pore loops on the uncapped side of ClpP act as swinging doors that open in one direction.

Other comments:

- Please include example trace fits for the ClpP and ClpX cryo-EM densities with fitted atomic coordinates in supplementary methods

- This sentence is not clear, consider rephrasing "thus the resulting density map of one ClpX hexamer represents a errant average of multiple conformations."

Ahmad Jomaa

We thank the reviewers for their comments and suggestions. We have performed the additional experiments requested, including expanding studies of the effects of an open-channel mutant on substrate specificity *in vitro* and including new experiments on mutant expression effects on cell physiology. For these studies we designed a new open-channel mutant containing a three-residue deletion in the ClpX pore-2 loop (Δ NPS), as the more drastic deletion/insertion used in our original manuscript bound ClpP less tightly than wild-type ClpP, which would have complicated studies *in vivo*, and also removed two pore-2 residues known to contact the *ssrA* tag in the 'recognition complex'. Destabilizing the closed-channel conformation of the pore-2 loop would still be expected to weaken or eliminate ClpX binding to the *ssrA* tag, and we find that Δ NPS ClpX ^{Δ N} or full-length Δ NPS ClpX support slow ClpP degradation of GFP-*ssrA* compared to otherwise identical enzymes with wild-type pore-2 loops *in vitro* and *in vivo*. We now also show that Δ NPS variants of ClpX ^{Δ N} or full-length ClpX support faster ClpP degradation of an unfolded protein lacking degrons (^{A^F}titin) as well as faster degradation of FITC-casein compared to parents with wild-type pore-2 loops. Expression of these Δ NPS variants in *E. coli* results in two phenotypes: (*i*) poorer survival in the presence of kanamycin; and (*ii*) slower cell growth at 42 °C. We have rewritten the paper extensively to emphasize our now better-characterized changes with respect to ClpXP substrate specificity. We also rewrote the section on ClpP peptide ingress/egress for improved clarity in response to reviewer comments.

REVIEWER COMMENTS

Reviewer #1 (Remarks to the Author):

The authors describe cryo-EM structures of the bacterial ClpXP ATP-dependent protease in the substrate-free state. These structures show that the axial pore of the ClpX ATPase is closed in the absence of substrate, despite the entry pore of ClpP compartmental protease being open and ready to accommodate substrate. Specifically, elements in the pore-2 loop of a ClpX subunit positioned low in the ATPase spiral blocks passage of substrates that are presumably displaced to initiate translocation. The authors note that the closed ClpX pore is inconsistent with previous experimental evidence showing nonspecific entry of a short reporter peptide into ClpP upon ClpX binding. Competition assays show that saturation of the specifically degraded *ssrA*-tagged substrates does not significantly impact peptide entry, suggesting alternative entry paths for the two types of substrates. Instead, the authors suggest that these peptides, and short peptide products produced during proteolysis, move via spaces created between the ClpX and ClpP domains.

The substrate-free ClpXP cryo-EM structures are a useful addition to the collection of physiologically relevant states of this protease and the observation that pore-loop 2 of ClpX blocks entry to ClpP in the absence of substrate provides a nice structural explanation for the tight regulation observed for this protease. The microscopy is well-performed and the structures will be of interest to the broader field of regulated proteolysis.

The question of how product peptides are released from compartmental proteases is important and understudied. However, the authors provide only a small amount of experimental evidence to support their model that peptides can be released between the ClpX and ClpP interface. A more detailed investigation is warranted to address the details of this model, such as the size of product that can be efficiently released and the coordination of the ATPase cycle with the appearance of the channels.

The size of nonspecific peptides that can enter or leave the ClpP chamber are similar. For example, ClpXP degradation products range from 6-28 amino acids, although the vast majority are 20 residues or smaller (Tremblay et al., 2020; Mawla et al., 2021). We now cite these references. ClpX stimulates degradation of peptides as large as 20 residues (Lee et al., 2010), and larger nonspecific peptides are likely to be able to enter ClpP more slowly. We suggested that peptide ingress/egress involves an open ClpP portal and a gap between ClpX IGF-loops spanning on open ClpP cleft, as these structural features are present in all high-resolution structures of ClpXP, including hydrolytically active ClpX variants with ATP or ATP γ S (which is hydrolyzed slowly by ClpX) and variants that can't hydrolyze ATP because of a Walker-B mutation. Thus, the IGF-gap and open portal are present in all highly populated stages of the ATPase cycle represented structurally. Both in cryo-EM structures of ClpXP and in many crystal structures of ClpP, alternative entry sites into or out of the ClpP chamber (e.g., equatorial windows) are not observed. We cannot eliminate the possibility that some peptides enter or leave the chamber via transient openings that are not observed structurally (Sprangers *et al.*, 2005) but believe that the IGF-gap/portal model is simplest and most likely.

Reviewer #2 (Remarks to the Author):

The bacterial AAA+ protease ClpX/ClpP degrades proteins harboring specific degrons like the SsrA-tag. Unspecific degradation of other proteins has to be prevented to avoid cellular toxicity. Here the authors present the cryo-EM structure of a substrate-free ClpX/ClpP complex. They show that the pore-2 loop of one subunit blocks the central translocation channel. The new structure represents an extension of former cryo-EM analysis by the authors. A similar structure was reported before in presence of substrate and was termed recognition complex, representing an initial step of substrate engagement. Deleting the pore-2 loop strongly enhanced degradation of a non-specific substrate (FITC-casein) by ClpX/ClpP, suggesting that channel closure contributes to substrate selectivity. This is an interesting finding that requires further support by controls as detailed below. Furthermore, it would be interesting to analyze the physiological consequences of such mutant on e.g. the ClpX/ClpP substrate spectrum.

The authors also suggest a new path for the release of peptides generated in the chamber of the associated ClpP peptidase barrel. Former studies (Sprangers et al., 2005) implied that peptide product release happens through the interface between the two heptameric ClpP rings. Accordingly, restricting ClpP mobility at this site was shown before to slow down product release

As Sprangers et al. (2005) demonstrated, a disulfide bond across the equatorial interface of ClpP dramatically slows release of a succinyl-Leu-Tyr-AMC dipeptide from the mutant chamber until the disulfide is reduced. However, this disulfide-bonded ClpP assumes an inactive compressed conformation (pdb 3HLN), and succinyl-Leu-Tyr-AMC enters wild-type ClpP at the same rate in the presence and absence of ClpX (Lee et al., 2010), making it a poor mimic for how ClpX alters normal ClpP peptide egress or ingress. Thus, we feel that our IGF-gap/portal model does a better job of explaining how degradation products leave the wild-type chamber of ClpXP and is consistent with all near-atomic resolution structures of ClpXP.

Here, the authors propose an alternative pathway, namely peptide entry and exit via the asymmetric interface between ClpX and ClpP. This model is indirectly supported by the finding that ClpX stimulates the degradation of a nonspecific decapeptide compared to peptide degradation by ClpP only even in the presence of the competing specific ClpX-substrate DHFR-SsrA.

The study comes up with two interesting mechanistic aspects, which are important for the understanding of AAA+ proteases. The biochemical support is, however, so far largely indirect and additional controls as detailed below are required to bolster the models. Furthermore, the physiological consequences of a ClpX mutant with reduced substrate selectivity should be explored. Such data could underline the relevance of channel closure for bacterial homeostasis. The study is therefore send back for major revision.

Major points:

Channel closure is also seen for the initial step of engagement of a specific SsrA-tagged substrate. This indicates that the binding of a specific substrate induces channel opening in a second step. Is it clear why only specific substrate but not unspecific ones trigger this step? The observation that a pore2-loop mutant degrades FITC-casein indicates that unspecific substrates can engage the upper part of the ClpX channel, yet without triggering channel opening.

Degron binding, by itself, is not known to induce channel opening. In fact, the channel is closed when the ssrA degron is specifically bound. Subsequent channel opening may require a power stroke and a substrate bound to the closed channel. By this model substrates that bind weakly to the closed-channel conformation are unlikely to be present when a power stroke occurs. Our data suggest that the closed and open channel conformations of ClpX are in an equilibrium that favors the closed state by a factor of ~10-fold. Thus, slow degradation of unfolded or poorly folded substrates by ClpX/ClpP or ClpX^{ΔN}/ClpP is probably mediated by a small population of open-channel enzymes.

Figure 4: The pore2-loop mutant was only analyzed in a ClpX deletion construct lacking the N-terminal domain. The same mutant needs to be analyzed in ClpX WT, as NTDs can function as selectivity barriers. Furthermore, the authors need to determine the proteolytic activity of the mutant towards a specific substrate, e.g. harboring an SsrA-tag. Unaltered

degradation of such substrate would document that the pore2-loop deletion only affects degradation of unspecific but not specific substrates, as predicted by the model.

We added the requested experiments of degradation of an *ssrA*-tagged substrate. However, because the closed channel forms the binding site for the *ssrA* tag, we note that our model strongly predicts that an 'open-channel' mutant would degrade *ssrA*-tagged proteins poorly. We designed a new open-channel mutant (Δ NPS) that interacts normally with ClpP, as the mutant used in the original manuscript bound ClpP poorly. In the revised manuscript, we show that the Δ NPS mutant in either full-length ClpX or ClpX ^{Δ N} backgrounds supports slow ClpP degradation of GFP-*ssrA* compared to otherwise identical enzymes with wild-type pore-2 loops. We also show that the Δ NPS mutation facilitates faster degradation of an unfolded protein lacking degrons (^AFtitin) as well as FITC-casein in both backgrounds.

The observation that a pore2-loop mutant exhibits reduced substrate selectivity raises the question whether its production causes toxicity in *E. coli* cells. A reduced cell growth or viability upon mutant expression would indicate a physiological need for the proposed mechanism.

To address this issue, we expressed our open-channel variants of ClpX ^{Δ N}/ClpP or ClpX/ClpP in *E. coli*. We find that expression of the mutant reduced cell viability in the presence of kanamycin and slows cell growth at 42 °C. No growth phenotypes were observed under non-stress conditions. The stress phenotypes we observe could result from failure of the mutants to degrade normal substrates or an ability of the mutants to degrade new substrates.

Figure 5: The authors need to document that DHFR-SsrA is degraded at normal rates in presence of unspecific peptide, supporting the idea that the two substrates use different ways to enter the ClpP degradation chamber. So far the authors only assume unaltered DHFR-SsrA degradation but that needs to be documented. The conclusion that the decapeptide does not travel through the ClpX translocation channel is therefore not demonstrated.

We performed the requested control. Addition of the nonspecific decapeptide did not slow ClpXP degradation of the DHFR substrate, as shown now in Fig. 6b.

It is also strongly recommended to use the generated pore2-loop mutant, which leads to enhanced FITC-casein degradation, in the peptide degradation assay. If the model of the authors is correct this ClpX mutant should not affect the rate of peptide degradation, as the peptides are predicted to not travel through the ClpX channel.

As requested, we tested the 'open-channel' Δ NPS mutant in the decapeptide cleavage assay (Fig. S14) and found similar rates of decapeptide cleavage in the presence or absence of the protein substrate.

Page 5, left column: ClpA/ClpP efficiently degrades casein, leading to the suggestion that the axial channel of ClpA is open in absence of substrates. This conclusion ignores the possibility that ClpA has a higher affinity to casein as compared to ClpX. A binding site for disordered proteins like casein has been described in the ClpB/Hsp104 N-terminal domain, which is homologous to ClpA. Thus differences in substrate binding but not channel opening/closure might underlie the diverse proteolytic activities.

As the reviewer suggests, differences in substrate binding almost certainly underlie most if not all of the phenotypes we observe *in vitro* and *in vivo*. For casein, we found that substantially enhanced degradation was observed for Δ NPS ClpX ^{Δ N}/ClpP but only marginally for Δ NPS ClpXP. This result is consistent with the reviewer's suggestion that the N-terminal domain may help bind casein. Nevertheless, it is striking that Δ NPS variants support faster ClpP degradation of unfolded ^{AF}titin compared to counterparts with wild-type pore-2 loops.

Reviewer #3 (Remarks to the Author):

This manuscript describes the substrate-free structure of ClpXP and reveals a closed conformation of the ClpX ring that was seen previously in the presence of substrate perched on the edge of the axial pore. This new structure verifies that the aforementioned substrate was indeed incumbent to an open pore and not involved in the pore restriction per se. It also confirms the mechanism of pore-loop2-mediated autoinhibition of ClpX-dependent access to ClpP. The authors go on to hypothesize that the non-ClpX-dependent substrates (such as decapeptides) may not travel through the ClpX pore, but instead travel through the gap left by the asymmetric pairing between hexameric ClpX and Heptameric ClpP at the ClpXP interface. While this hypothesis has merit, we could use a few details.

1.How big is the IGF opening within the static EM structure?

The opening is ~10 Å, which we now cite.

2.How do the interactions of the ClpX-pore2 and ClpP-N-terminal loops play into this trans axial access? Figure 1 of reference #24 suggests these loop interactions if not block exterior access, certainly funnel, peptides from ClpX to ClpP. How is trans-axial access impeded or facilitated by these loop interactions?

The structure in Fig. 1 of reference 24 wasn't 'real', as no high-resolution structures of ClpX or ClpXP were known at that time. Rather, it was based on modeling using a crystal structure of ClpP and a crystal structure of another AAA+ protease (HslUV), which doesn't have the 6-7 symmetry mismatch of ClpXP and differs in many other ways. In cryo-EM structures of ClpXP, we and others see some interactions between the β -hairpins of ClpP and the pore-2 loops of ClpX, but these interactions are not extensive.

3.If the pore2 loops make protein:protein contacts, then the increase in activity in figure 4 would be from both release of axial and trans-axial blockage?

Good point. In the original manuscript we used a pore-2 mutant that weakened ClpX binding to ClpP. In the revised manuscript, we designed a new pore-2 mutant (Δ NPS) that binds ClpP well.

4.How is this hypothesis effected by the knowledge that related AAA+ ATPase unfoldases can harbor more than one peptide through the pore? (Lee et al. JBC (2002) and Han eLife (2019))

ClpX also translocates multiple polypeptides through the channel when substrates contain disulfide bonds (Burton et al., *EMBO J* 2001; Bolon et al., *Mol Cell* 2004). Thus, we can't rigorously exclude the possibility that nonspecific peptides enter ClpP and peptide products leave ClpP through the axial channel of ClpX, even when it is filled with a substrate polypeptide. However, this model seems unlikely, in part because of the closed axial channel in substrate-free ClpX and also because nonspecific peptides can enter the degradation chamber of ClpXP even when ClpX can't translocate polypeptides because ATP hydrolysis is compromised using ATP γ S and a Walker-B ATPase mutation (Lee et al., *JMB* 2010). Under these conditions, peptides or polypeptides are trapped in the ClpX channel, as we observed in our first cryo-EM structures of ClpXP (Fei, Bell, Jenni *et al.*, eLife, 2020).

It is interesting to note that the methods say the "substrate-free" structure was obtained in the presence of substrate λ O-Arc – not observed. While unfortunate luck and heterogeneous analysis of CryoEM particles can easily provide rationale for the substrate-free structure, it seems this tidbit of information is somewhat hidden in the methods details.

Our attempt to bury this tidbit clearly failed. Suitably chastised, we now cite this fact upfront in the main text.

1.Should details be provided as to the peptide sequence or reference to its use as a substrate?

The sequence of the peptide (Abz-KASPVSLGY^{NO2}D) was listed in the Methods along with the reference (Lee et al., *JMB* 2010).

The protein preparation, CryoEM SP data collection, processing and model refinement all look appropriate for this work.

Reviewer
Heidi L Schubert

Reviewer #4 (Remarks to the Author):

Ghanbarpour et al. addresses an important question in the regulation of the ClpXP protease, which confers specificity for substrate degradation: Is the ClpX channel opened or closed prior to the binding and translocation of a substrate? In this work, the authors addressed this elegantly with a series of cryo-EM and biochemical experiments. First, they determined cryo-EM structures of a substrate-free full-length and an N-terminally truncated ClpX complexed with ClpP. The cryo-EM structures revealed an open axial ClpP channel and a closed ClpX channel. Second, the authors showed that mutations in the pore loop of ClpX de-regulate ClpP's substrate specificity and degradation. Finally, through biochemical experiments, the authors propose a model for short peptides' entry into the degradation chamber of ClpP through the gap generated by the symmetry mismatch at the interface between ClpX and ClpP.

The manuscript is well-written and addresses a fundamental pathway on how AAA+ proteases carry out degradation of substrates; in particular how ClpXP regulates substrate entry into the degradation chamber. The model is well-supported by the set of experiments performed here. I only have minor suggestions/questions before the publication of this manuscript.

Comments:

- On page 2, the flexible ~60 residues in the N-terminal region of ClpX is mentioned in the text with reference to supplementary figures (Fig S4 and S5). It is not clear where this region would be in the structure. A dashed line or perhaps a schematic would help.

The ClpX N-terminal domain is a homodimer of subunits ~51 residues in length. It is flexibly connected to the rest of ClpX by a unstructured linker of ~15 amino acids. Based on the points of attachment the N-terminal dimers are likely to be on the periphery of the ClpX hexamer, which we have attempted to illustrate by the dotted lines in the revised Fig. S1b.

- In the EM images of the full length ClpXP complex, there seem to be 'some' stacking of ClpXP particles (Fig S2). This is not the case in the images with the N-terminally truncated ClpX mutant (Fig S1). Perhaps it is not specific, but did the authors perform focused 3D classification/variability analysis to rule out the possibility that this mediated via N-terminal interactions as previously reported? The box size used here is small, maybe refining/classifying with a larger box size?

To address this issue, we extracted particles from the final refinement using a box that was large enough to include the N-terminal domains (800 pix Fourier cropped to 440 pix). A fresh reconstruction was performed with the newly extracted particles. The reconstruction was checked for N-terminal-domain density. As shown in the figure below, the normal and permissive isosurface levels did not show the presence of N-terminal domains. To ensure that ClpXP did not dominate the alignment and hinder the visualization of N-domain, we centered particles around ClpX in the extraction box

using the 'Volume alignment tools', and a mask was created using this map. This mask was then used to perform particle-signal subtraction. A fresh reconstruction was performed using *ab-initio* reconstruction and homogeneous refinement from the signal-subtracted particles. As shown in the lower portion of the figure below, this reconstruction was very noisy and provided no evidence for the presence of N-terminal domains in fixed conformations relative to the AAA+ module of ClpX.

- CryDrgn2 was used to investigate whether the ClpX channel could be opened by stochastic ATP hydrolysis in the substrate free form. This software is a powerful tool and the results for this particular sample could be shown in the supplementary. Please include some examples of the obtained 3D volumes, in particular the opened channel state.

We now show these volumes in Fig. S9b.

- The authors propose a model for small peptides' entry through gaps generated by unoccupied pockets due to symmetry mismatch. This model is plausible and would indicate that the power stroke generated by ATP hydrolysis in ClpX is not required for translocation of small peptides. To validate this, did the authors test peptide degradation in the presence of ATPγS or ADP, where power strokes are inhibited?

In this paper, we do show that the decapeptide is degraded robustly by E185Q ClpX^{ΔN}/ClpP which hydrolyzes ATP very slowly as a consequence of a Walker-B mutation. In Lee *et al.* (2010), we also showed that the decapeptide was degraded at essentially the wild-type rate using ATPγS and E185Q ClpXP. ADP doesn't support binding of ClpX to ClpP, so the outcome of that experiment can't be interpreted.

- In the text, it is indicated that two of the ClpX protomers were in the ADP state in the cryo-EM maps. Are these the two subunits that break the 6-fold symmetry of the hexamer i.e., as observed in the recognition complex? It would be easier to assess this better with corresponding EM densities/coordinates for the bound ATP/ADP molecules in the supplementary materials, showing closeups of the conformational changes in the interface between ClpX protomers.

The subunits of ClpX are arranged in a shallow spiral and don't have 6-fold rotational symmetry in any known structures.

- A schematic for the proposed model addressing the two-step substrate recognition/engagement would be helpful for the readership.

We have added the requested model as Fig. S8. Movie S1 also shows our proposed two-step model.

- Could the authors comment on the ClpX concentration in the cell? Is it higher than ClpP? One would assume that ClpP is more abundant and thus ClpXP is likely to be a single-capped complex in the cell.

There are ~100 ClpX₆ molecules per cell (Farrell *et al.*, *Mol Micro* 2005). The ClpP₁₄ concentration is ~100 per cell during logarithmic growth but increases to ~300 per cell during stationary phase. However, ClpA₆ (50 to 125 molecules per cell depending on growth conditions) also binds ClpP₁₄ and can form ClpA₆•ClpP₁₄•ClpX₆ complexes. This is a long-winded way of saying that it's hard to answer the singly versus doubly capped question in cells. In the test tube, however, it's easy to ensure that almost all ClpXP complexes are doubly capped, simply by added a large excess of ClpX at a concentration above the K_D for ClpX-ClpP binding. Under these conditions, nonspecific peptides still enter ClpP and steady-state protein degradation, which requires product release, is not compromised.

- It is difficult to envision how peptide products, to access the equatorial ring-ring gaps, will have to first pass through the same axial pore of ClpP as substrate is being translocated. This is a tight space and would likely affect the rate of degradation. Is it

plausible that these peptides exit ClpP through the uncapped end instead? i.e., the axial pore loops on the uncapped side of ClpP act as swinging doors that open in one direction.

When complexed with ClpX, the axial portal of ClpP has a diameter of ~ 30 Å, leaving more than enough room for an incoming translocated polypeptide and outgoing peptide products of degradation. We certainly can't exclude the possibility that some peptide products in singly capped ClpXP leave through the uncapped end, although it appears closed in our structure, but this isn't an option in doubly capped ClpXP. The one-way swinging door model is interesting, but we're not sure what physical principles would enforce directionality and preclude microscopic reversibility. Moreover, if it's truly one-way, then nonspecific peptides would have to enter singly capped ClpXP by a different route. Our model, by contrast, is simpler, accounts for singly and doubly capped ClpXP activities, and allows nonspecific peptides to enter the degradation chamber by the same route that peptide products of degradation normally leave.

Other comments:

- Please include example trace fits for the ClpP and ClpX cryo-EM densities with fitted atomic coordinates in supplementary methods

These fits are now shown in Fig. S7.

- This sentence is not clear, consider rephrasing "thus the resulting density map of one ClpX hexamer represents a errant average of multiple conformations."

We rewrote this sentence to read 'For the doubly capped ClpXP structures, only one of the two ClpX rings was included in the refinement because the conformations of the two ClpX hexamers are uncoordinated and thus one has poor resolution because of structural averaging.'

Ahmad Jomaa

REVIEWER COMMENTS

Reviewer #1 (Remarks to the Author):

The manuscript is improved by the addition of the *in vivo* experiments demonstrating the importance of the pore-2 loop residues to survival under stress conditions. These experiments and additional controls requested by the other reviewers support the authors conclusions and have been performed satisfactorily. The evidence supporting the model for exit and entry of peptides is still largely indirect. However, this would be technically challenging to demonstrate directly and the data currently presented merit publication.

Reviewer #2 (Remarks to the Author):

In their revised version the authors have largely addressed all my previous concerns. The clarifications on how the pore2-loops indirectly contribute to substrate specificity are valuable and important. Furthermore, new findings support an alternative peptide path into the ClpP chamber that is independent from the translocation channel of interacting ClpX. A limitation remains the *in vivo* relevance of enhanced degradation of unstructured proteins by ClpX pore2-loop mutants. The undertaken efforts are appreciated, the provided data however rather support a loss but not a gain of degradation activity by the mutant. The physiological relevance therefore remains unanswered and the author`s conclusion on the importance of channel closure (see abstract) does not seem justified. The authors are asked to revise this point accordingly.

Product release model:

The reviewer is not completely following the rationale of the authors about previous work by Houry and colleagues. Isn`t the observation that stabilization of the compressed, inactive form of ClpP by disulfide bond formation is slowing down product release supporting the model that cleaved peptides leave ClpP via the equatorial interface? ClpP when bound to ClpX will be in the active extended conformation, which could thus allow for peptide release via the ClpP ring interface. While this reviewer is appreciating that peptides can enter ClpX-complexed ClpP directly (without passing through the ClpX translocation channel), this finding does not rule an alternative path for peptide exit.

In vivo relevance of pore2-loop:

Fig. 5b: The authors should provide quantifications of the cell filamentation phenotype.

Fig. 5c: The addition of growth curves and spot tests are appreciated. They however do not provide evidence that deregulation of ClpX pore2-loop mutants (leading to more efficient degradation on non-structured proteins) causes a growth defect. It is not possible to differentiate between a gain and loss of function phenotype as clpX mutant cells show the same growth defect.

Fig. 5d: The spot tests are not convincing. Plating efficiencies of cells harboring the diverse plasmids are identical at 10⁻⁵ dilution, so why should they become different at 10⁻⁶ (upper spot test)? Quantifications are also missing. The lower spot test is not clearly defined. Which conditions have been tested here? Overall the added *in vivo* data support a reduced activity of pore2-loop mutants (e.g. against SsrA-tagged substrates) but not enhanced activities due to relaxed/altered substrate specificity.

Reviewer #3 (Remarks to the Author):

Review of the revised version of "A closed translocation channel in the substrate-free AAA+ ClpXP protease diminishes rogue degradation."

This manuscript describes a substrate-free ClpXP CryoEM structure and experiments to test the role of residues which are responsible for channel closure. The structure is similar to known early SsrA-recognition structures reinforcing the concept that the axial pore closure is a default regulatory mechanism to regulate ClpXP selectivity. The authors go on to remove three residues of the pore-blocking loop and test this theory. A complicating factor of this test is that the NPS sequence not only forms a pore blocking loop in subunit A, but it also forms a ClpP-entry blocking loop in subunit E and perhaps F. The removal of 3-6 residues from the ClpP entry loop is likely significant, though not essential, for non-channel ClpP access.

Removal of pore-loop-2 residues involved in loop closure renders the enzyme effectively inactive against folded substrates (Fig 4a). This has more to do with loss of the "paddles" that the pore loops form to push substrates through the axial loop than the loss of loop closure. ATP hydrolysis fuels these conformational changes. This role of pore-loop 2 is not highlighted in the paper and a -ATP control would probably show that the ClpXdN/ClpP is sensitive to ATP where the dNPS is less so.

Unfolded substrates are tested in panels 4c and 4d and results are harder to interpret. What I think the authors want to show is that the NPS mutant relieves blockage to unfolded substrates – but because of the dual role of the pore-loop-2, it remains unclear if this is due to additional channel access.

Hence while it is true that the "NPS destabilizes the closed-channel conformation," it is unclear if that is the primary effect.

Reviewer #4 (Remarks to the Author):

The authors have adequately addressed all my comments.

REVIEWER COMMENTS

Reviewer #1 (Remarks to the Author):

The manuscript is improved by the addition of the in vivo experiments demonstrating the importance of the pore-2 loop residues to survival under stress conditions. These experiments and additional controls requested by the other reviewers support the authors conclusions and have been performed satisfactorily. The evidence supporting the model for exit and entry of peptides is still largely indirect. However, this would be technically challenging to demonstrate directly and the data currently presented merit publication.

We agree and thank reviewer #1.

Reviewer #2 (Remarks to the Author):

In their revised version the authors have largely addressed all my previous concerns. The clarifications on how the pore2-loops indirectly contribute to substrate specificity are valuable and important. Furthermore, new findings support an alternative peptide path into the ClpP chamber that is independent from the translocation channel of interacting ClpX. A limitation remains the in vivo relevance of enhanced degradation of unstructured proteins by ClpX pore2-loop mutants. The undertaken efforts are appreciated, the provided data however rather support a loss but not a gain of degradation activity by the mutant. The physiological relevance therefore remains unanswered and the author's conclusion on the importance of channel closure (see abstract) does not seem justified. The authors are asked to revise this point accordingly.

The reviewer is correct that we haven't demonstrated a gain of degradation activity by the mutant *in vivo*. Moreover, this would be very difficult to do given the large number of natural ClpXP substrate and the possibility of new substrates based on our experiments *in vitro* that clearly show enhanced degradation of unfolded and poorly folded substrates lacking degrons by the open-channel mutant. To address, the reviewer's concern, we removed physiological relevance from the last sentence of the Abstract, which now reads '*Thus, channel closure is an important determinant of ClpXP degradation specificity.*'

Product release model:

The reviewer is not completely following the rationale of the authors about previous work by Houry and colleagues. Isn't the observation that stabilization of the compressed, inactive form of ClpP by disulfide bond formation is slowing down product release supporting the model that cleaved peptides leave ClpP via the equatorial interface? ClpP when bound to ClpX will be in the active extended conformation, which could thus allow for peptide release via the ClpP ring interface. While this reviewer is appreciating that

peptides can enter ClpX-complexed ClpP directly (without passing through the ClpX translocation channel), this finding does not rule an alternative path for peptide exit.

We can't and don't rule out the 'equatorial' model of Houry and colleagues, which we describe and cite in one sentence on page 15 of the text. In a following sentence, we point out that equatorial windows have not been observed structurally, whereas the IGF gap is a feature of all known ClpXP structures. We are happy to allow readers to weigh the merits of both models and to reach their own conclusions and note that our entire discussion of the Houry model is limited to these two sentences. Thus, this issue is a very a minor focus of the paper.

In vivo relevance of pore2-loop:

Fig. 5b: The authors should provide quantifications of the cell filamentation phenotype.

As requested, we have quantified the cell-filamentation phenotype (see below) and have included this graph as part of Fig. 5b in the revised manuscript.

Fig. 5c: The addition of growth curves and spot tests are appreciated. They however do not provide evidence that deregulation of ClpX pore2-loop mutants (leading to more efficient degradation on non-structured proteins) causes a growth defect. It is not possible to differentiate between a gain and loss of function phenotype as clpX mutant cells show the same growth defect.

We agree with the reviewer and indeed had stated on page 10 'In principle, expression of ΔNPS ClpX^{ΔN} or ΔNPS ClpX in *E. coli* might affect cell physiology by failing to degrade normal ClpXP substrates, such as ssrA-tagged proteins, and/or by degrading new substrates.'

Fig. 5d: The spot tests are not convincing. Plating efficiencies of cells harboring the diverse plasmids are identical at 10⁻⁵ dilution, so why should they become different at 10⁻⁶ (upper spot test)? Quantifications are also missing. The lower spot test is not clearly defined. Which conditions have been tested here?

We replaced the spot tests with direct colony counts in plating assays following dilution of cultures to allow quantification. The new data is shown in Fig. 5d.

Overall the added in vivo data support a reduced activity of pore2-loop mutants (e.g. against SsrA-tagged substrates) but not enhanced activities due to relaxed/altered substrate specificity.

As discussed above, we agree and made this point explicitly on page 10.

Reviewer #3 (Remarks to the Author):

Review of the revised version of “A closed translocation channel in the substrate-free AAA+ ClpXP protease diminishes rogue degradation.”

This manuscript describes a substrate-free ClpXP CryoEM structure and experiments to test the role of residues which are responsible for channel closure. The structure is similar to known early SsrA-recognition structures reinforcing the concept that the axial pore closure is a default regulatory mechanism to regulate ClpXP selectivity. The authors go on to remove three residues of the pore-blocking loop and test this theory. A complicating factor of this test is that the NPS sequence not only forms a pore blocking loop in subunit A, but it also forms a ClpP-entry blocking loop in subunit E and perhaps F. The removal of 3-6 residues from the ClpP entry loop is likely significant, though not essential, for non-channel ClpP access.

We thank reviewer #3 for their comments. We are somewhat confused, however, about the model that ‘the NPS sequence not only forms a pore blocking loop in subunit A, but it also forms a ClpP-entry blocking loop in subunit E and perhaps F.’ In none of our cryo-EM structures does the pore-2 loop of ClpX subunits E or F loop block entry into ClpP. We know of no other cryo-EM structures or experiments that support this idea.

Removal of pore-loop-2 residues involved in loop closure renders the enzyme effectively inactive against folded substrates (Fig 4a).

Δ NPS ClpXP does degrade GFP-ssrA but more slowly than the wild-type enzyme. Given that GFP-ssrA unfolds spontaneously with a half-life of ~ 20 years, the degradation observed indicates that the mutant enzyme can unfold very stable proteins. To establish that the reduced rate is at least partially a consequence of a reduced K_M for degradation, we measured rates at different substrate concentrations. This data is now included as a revised Fig. 4a in the new manuscript.

These data show a K_M of $\sim 2 \mu\text{M}$ for degradation of GFP by ClpX $^{\Delta$ N/ClpP but a K_M greater than $15 \mu\text{M}$ for Δ NPS ClpX $^{\Delta$ N/ClpP. Because there is no significant curvature in the Δ NPS ClpX $^{\Delta$ N/ClpP data, it is not possible to determine K_M and V_{max} values via Michaelis-Menten analysis, but a linear dependence of the enzymatic rate on substrate concentration is only expected at substrate concentrations substantially below K_M . At $15 \mu\text{M}$ GFP-ssrA*, the Δ NPS variant degrades the substrate at $\sim 30\%$ of the wild-type rate, and this value would increase at higher substrate concentrations. Thus, we disagree that the Δ NPS deletion ‘renders the enzyme effectively inactive against folded substrates.’ Moreover, Δ NPS ClpXP has cellular phenotypes similar to wild-type ClpXP in terms of phage Mu plating, which requires remodeling of a very stable complex of the MuA transposase bound to DNA (this complex is stable in 6 M urea for hours), and cell filamentation, which requires degradation of the stable FtsZ protein. Again, these results do not support a major defect in protein unfolding/remodeling for Δ NPS ClpXP.

This has more to do with loss of the “paddles” that the pore loops form to push substrates through the axial loop than the loss of loop closure. ATP hydrolysis fuels these conformational changes. This role of pore-loop 2 is not highlighted in the paper and a -ATP control would probably show that the ClpXdN/ClpP is sensitive to ATP where the dNPS is less so.

The ‘pore-loop paddles’ are required for ATP-dependent translocation and for substrate unfolding. Because Δ NPS ClpX/ClpP degrades some poorly folded

substrates faster than the wild-type parent, the mutant paddles don't seem to be deficient in translocation. As noted above, the mutant paddles are also capable of unfolding GFP, a very stable native substrate. To test the reviewer's prediction about ATP dependence, we assayed degradation of FITC-casein in the presence of ATP or ATP γ S, which ClpX hydrolyzes more slowly than ATP. As shown below (**Fig. S11** in the revised manuscript), degradation of FITC-casein is strongly dependent on the rate of ATP hydrolysis, a result in conflict with the reviewer's prediction.

Unfolded substrates are tested in panels 4c and 4d and results are harder to interpret. What I think the authors want to show is that the NPS mutant relieves blockage to unfolded substrates – but because of the dual role of the pore-loop-2, it remains unclear if this is due to additional channel access. Hence while it is true that the “ Δ NPS destabilizes the closed-channel conformation,” it is unclear if that is the primary effect.

We reiterate that enhanced degradation of unfolded substrates by Δ NPS ClpXP remains dependent on the rate of ATP hydrolysis. Furthermore, we emphasize that the pore-2 loop of chains E and F, as demonstrated by cryo-EM structures from our group and other researchers, does not form a blockage for ClpP entry. Our biochemical results support a model in which the Δ NPS mutation allows unfolded substrates to better access the axial channel and that it destabilizes the closed-channel binding site for the ssaA tag. This model also makes sense in terms of the cryo-EM structures available. Again, we are unsure what alternative model the reviewer is suggesting. A model in which the Δ NPS mutation only effects the mechanical substrate-processing properties of ClpX is inconsistent with the increased K_M for degradation of GFP-ssaA*. We cannot eliminate the possibility that changes in the pore-2 loop also effect rates of unfolding/translocation (V_{max}). However, Δ NPS ClpX Δ N/ClpP degrades unfolded titin with or without an ssaA tag at comparable rates, whereas ClpX Δ N/ClpP degrades these substrates at rates differing by 10-fold or more. These results support the idea that the major effect of the Δ NPS mutation is to affect substrate recognition and not the rate of

substrate processing. This interpretation is also supported by our finding that the Δ NPS deletion increases K_M for degradation of GFP-ssrA*, but is still capable of degrading this highly stable native substrate.

Reviewer #4 (Remarks to the Author):

The authors have adequately addressed all my comments.

We thank reviewer #4.

REVIEWERS' COMMENTS

Reviewer #2 (Remarks to the Author):

In their newly revised version the authors have addressed all my former concerns. Added clarifications are appreciated and publication of the study in Nature Communications is justified.

Reviewer #3 (Remarks to the Author):

This reviewer (#3) apologizes if they misinterpreted/overstated the role of the NPS loop in ClpX-ClpP interactions at the pore-to-pore interface. I was thinking that the pore loops of lower subunits were involved in that interface and hence limited non-ClpX-translocating substrates - the unfolded ones being tested. I am willing to accept the arguments presented in response to my earlier comments. I did appreciate the ATP-dependence graph. thank you.